# Learning Diffusion Priors from Observations by Expectation Maximization

**François Rozet**
University of Liège
francois.rozet@uliege.be

**Gérôme Andry**
University of Liège
gandry@uliege.be

**François Lanusse**
Université Paris-Saclay,
Université Paris Cité, CEA, CNRS, AIM
francois.lanusse@cnrs.fr

**Gilles Louppe**
University of Liège
g.louppe@uliege.be

## Abstract

Diffusion models recently proved to be remarkable priors for Bayesian inverse problems. However, training these models typically requires access to large amounts of clean data, which could prove difficult in some settings. In this work, we present a novel method based on the expectation-maximization algorithm for training diffusion models from incomplete and noisy observations only. Unlike previous works, our method leads to proper diffusion models, which is crucial for downstream tasks. As part of our method, we propose and motivate an improved posterior sampling scheme for unconditional diffusion models. We present empirical evidence supporting the effectiveness of our method.

## 1 Introduction

Many scientific applications can be formalized as Bayesian inference in latent variable models, where the target is the posterior distribution $p(x \mid y) \propto p(y \mid x) \, p(x)$ given an observation $y \in \mathbb{R}^M$ resulting from a forward process $p(y \mid x)$ and a prior distribution $p(x)$ over the latent variable $x \in \mathbb{R}^N$. Notable examples include gravitational lensing inversion [1–3], accelerated MRI [4–8], unfolding in particle physics [9, 10], and data assimilation [11–14]. In all of these examples, the observation $y$ alone is either too incomplete or too noisy to recover the latent $x$. Additional knowledge in the form of an informative prior $p(x)$ is crucial for valuable inference.

Recently, diffusion models [15, 16] proved to be remarkable priors for Bayesian inference, demonstrating both quality and versatility [17–27]. However, to train a diffusion model for the latent $x$, one would typically need a large number of latent realizations, which by definition are not or rarely accessible. This is notably the case in earth and space sciences where the systems of interest can only be probed superficially.

Empirical Bayes (EB) methods [28–31] offer a solution to the problem of prior specification in latent variable models when only observations $y$ are available. The objective of EB is to find the parameters $\theta$ of a prior model $q_\theta(x)$ for which the evidence distribution $q_\theta(y) = \int p(y \mid x) \, q_\theta(x) \, dx$ is closest to the empirical distribution of observations $p(y)$. Many EB methods have been proposed over the years, but they remain limited to low-dimensional settings [32–37] or simple parametric models [38, 39].

In this work, our goal is to use diffusion models for the prior $q_\theta(x)$, as they are best-in-class for modeling high-dimensional distributions and enable many downstream tasks, including Bayesian inference. This presents challenges for previous empirical Bayes methods which typically rely on models for which the density $q_\theta(x)$ or samples $x \sim q_\theta(x)$ are differentiable with respect to the

38th Conference on Neural Information Processing Systems (NeurIPS 2024).

parameters $\theta$. Instead, we propose an adaptation of the expectation-maximization [40–44] algorithm where we alternate between generating samples from the posterior $q_\theta(x \mid y)$ and training the prior $q_\theta(x)$ on these samples. As part of our method, we propose an improved posterior sampling scheme for unconditional diffusion models, which we motivate theoretically and empirically.

## 2 Diffusion Models

The primary purpose of diffusion models (DMs) [15, 16], also known as score-based generative models [45, 46], is to generate plausible data from a distribution $p(x)$ of interest. Formally, adapting the continuous-time formulation of Song et al. [46], samples $x \in \mathbb{R}^N$ from $p(x)$ are progressively perturbed through a diffusion process expressed as a stochastic differential equation (SDE)

$$\mathrm{d}x_t = f_t\, x_t\, \mathrm{d}t + g_t\, \mathrm{d}w_t \tag{1}$$

where $f_t \in \mathbb{R}$ is the drift coefficient, $g_t \in \mathbb{R}_+$ is the diffusion coefficient, $w_t \in \mathbb{R}^N$ denotes a standard Wiener process and $x_t \in \mathbb{R}^N$ is the perturbed sample at time $t \in [0, 1]$. Because the SDE is linear with respect to $x_t$, the perturbation kernel from $x$ to $x_t$ is Gaussian and takes the form

$$p(x_t \mid x) = \mathcal{N}(x_t \mid \alpha_t\, x, \Sigma_t) \tag{2}$$

where $\alpha_t$ and $\Sigma_t = \sigma_t^2 I$ are derived from $f_t$ and $g_t$ [46–49]. Crucially, the forward SDE (1) has an associated family of reverse SDEs [46–49]

$$\mathrm{d}x_t = \left[ f_t\, x_t - \frac{1+\eta^2}{2} g_t^2\, \nabla_{x_t} \log p(x_t) \right] \mathrm{d}t + \eta\, g_t\, \mathrm{d}w_t \tag{3}$$

where $\eta \geq 0$ is a parameter controlling stochasticity. In other words, we can draw noise samples $x_1 \sim p(x_1) \approx \mathcal{N}(0, \Sigma_1)$ and gradually remove the noise therein to obtain data samples $x_0 \sim p(x_0) \approx p(x)$ by simulating Eq. (3) from $t = 1$ to $0$ using an appropriate discretization scheme [16, 45, 46, 49–52]. In this work, we adopt the variance exploding SDE [45] for which $f_t = 0$ and $\alpha_t = 1$.

In practice, the score function $\nabla_{x_t} \log p(x_t)$ in Eq. (3) is unknown, but can be approximated by a neural network trained via denoising score matching [53, 54]. Several equivalent parameterizations and objectives have been proposed for this task [16, 45, 46, 50–52]. In this work, we adopt the denoiser parameterization $d_\theta(x_t, t)$ and its objective [51]

$$\arg \min_\theta \mathbb{E}_{p(x)p(t)p(x_t \mid x)} \left[ \lambda_t \left\| d_\theta(x_t, t) - x \right\|_2^2 \right], \tag{4}$$

for which the optimal denoiser is the mean $\mathbb{E}[x \mid x_t]$ of $p(x \mid x_t)$. Importantly, $\mathbb{E}[x \mid x_t]$ is linked to the score function through Tweedie's formula [55–58]

$$\mathbb{E}[x \mid x_t] = x_t + \Sigma_t \nabla_{x_t} \log p(x_t), \tag{5}$$

which allows to use $s_\theta(x_t) = \Sigma_t^{-1}(d_\theta(x_t, t) - x_t)$ as a score estimate in Eq. (3).

## 3 Expectation-Maximization

The objective of the expectation-maximization (EM) algorithm [40–44] is to find the parameters $\theta$ of a latent variable model $q_\theta(x, y)$ that maximize the log-evidence $\log q_\theta(y)$ of an observation $y$. For a distribution of observations $p(y)$, the objective is to maximize the expected log-evidence [43, 44] or, equivalently, to minimize the Kullback-Leibler (KL) divergence between $p(y)$ and $q_\theta(y)$. That is,

$$\theta^* = \arg \max_\theta \mathbb{E}_{p(y)} \left[ \log q_\theta(y) \right] \tag{6}$$

$$= \arg \min_\theta \mathrm{KL}\big(p(y) \,\|\, q_\theta(y)\big). \tag{7}$$

The key idea behind the EM algorithm is that for any two sets of parameters $\theta_a$ and $\theta_b$, we have

$$\log \frac{q_{\theta_a}(y)}{q_{\theta_b}(y)} = \log \frac{q_{\theta_a}(x, y)}{q_{\theta_b}(x, y)} + \log \frac{q_{\theta_b}(x \mid y)}{q_{\theta_a}(x \mid y)} \tag{8}$$

$$= \mathbb{E}_{q_{\theta_b}(x \mid y)} \left[ \log \frac{q_{\theta_a}(x, y)}{q_{\theta_b}(x, y)} \right] + \mathrm{KL}\big(q_{\theta_b}(x \mid y) \,\|\, q_{\theta_a}(x \mid y)\big) \tag{9}$$

$$\geq \mathbb{E}_{q_{\theta_b}(x \mid y)} \left[ \log q_{\theta_a}(x, y) - \log q_{\theta_b}(x, y) \right]. \tag{10}$$

This inequality also holds in expectation over $p(y)$. Therefore, starting from arbitrary parameters $\theta_0$, the EM update

$$\theta_{k+1} = \arg\max_\theta \mathbb{E}_{p(y)} \mathbb{E}_{q_{\theta_k}(x|y)} \big[ \log q_\theta(x, y) - \log q_{\theta_k}(x, y) \big] \tag{11}$$

$$= \arg\max_\theta \mathbb{E}_{p(y)} \mathbb{E}_{q_{\theta_k}(x|y)} \big[ \log q_\theta(x, y) \big] \tag{12}$$

leads to a sequence of parameters $\theta_k$ for which the expected log-evidence $\mathbb{E}_{p(y)} \big[ \log q_{\theta_k}(y) \big]$ is monotonically increasing and converges to a local optimum [42–44].

When the expectation in Eq. (12) is intractable, many have proposed to use Monte Carlo approximations instead [59–66]. Previous approaches include Markov chain Monte Carlo (MCMC) sampling, importance sampling, rejection sampling and their variations [63–66]. A perhaps surprising advantage of Monte Carlo EM (MCEM) algorithms is that they may be able to overcome local optimum traps [60, 61]. We refer the reader to Ruth [66] for a recent review of MCEM algorithms.

## 4 Methods

Although rarely mentioned in the literature, the expectation-maximization algorithm is a possible solution to the empirical Bayes problem. Indeed, both have the same objective: minimizing the KL between the empirical distribution of observations $p(y)$ and the evidence $q_\theta(y)$. In the empirical Bayes setting, the forward model $p(y \mid x)$ is known and only the parameters of the prior $q_\theta(x)$ should be optimized. In this case, Eq. (12) becomes

$$\theta_{k+1} = \arg\max_\theta \mathbb{E}_{p(y)} \mathbb{E}_{q_{\theta_k}(x|y)} \big[ \log q_\theta(x) + \log p(y \mid x) \big] \tag{13}$$

$$= \arg\max_\theta \mathbb{E}_{p(y)} \mathbb{E}_{q_{\theta_k}(x|y)} \big[ \log q_\theta(x) \big] \tag{14}$$

$$= \arg\min_\theta \mathrm{KL}\big( \pi_k(x) \, \| \, q_\theta(x) \big) \tag{15}$$

where $\pi_k(x) = \int q_{\theta_k}(x \mid y) \, p(y) \, \mathrm{d}y$. Intuitively, $\pi_k(x)$ and therefore $q_{\theta_{k+1}}(x)$ assign more density to latents $x$ which are consistent with observations $y \sim p(y)$ than $q_{\theta_k}(x)$. In this work, we consider a special case of the empirical Bayes problem where each observation $y$ has an associated measurement matrix $A$ and the forward process takes a linear Gaussian form $p(y \mid x, A) = \mathcal{N}(y \mid Ax, \Sigma_y)$. This formulation allows the forward process to be potentially different for each observation $y$. For example, if the position or environment of a sensor changes, the measurement matrix $A$ may also change, which leads to an empirical distribution of pairs $(y, A) \sim p(y, A)$. As a result, $\pi_k(x)$ in Eq. (15) becomes $\pi_k(x) = \int q_{\theta_k}(x \mid y, A) \, p(y, A) \, \mathrm{d}y$.

### 4.1 Pipeline

Now that our goals and assumptions are set, we present our method to learn a diffusion model $q_\theta(x)$ for the latent $x$ from observations $y$ by expectation-maximization. The idea is to decompose Eq. (15) into (i) generating a dataset of i.i.d. samples from $\pi_k(x)$ and (ii) training $q_{\theta_{k+1}}(x)$ to reproduce the generated dataset. We summarize the pipeline in Algorithms 1, 2 and 3, provided in Appendix A due to space constraints.

**Expectation**  To draw from $\pi_k(x)$, we first sample a pair $(y, A) \sim p(y, A)$ and then generate $x \sim q_{\theta_k}(x \mid y, A)$ from the posterior. Unlike previous MCEM algorithms that rely on expensive and hard to tune sampling strategies [63–66], the use of a diffusion model enables efficient and embarrassingly parallelizable posterior sampling [21–23]. However, the quality of posterior samples is critical for the EM algorithm to converge properly [63–66] and, in this regard, we find previous posterior sampling methods [21–23, 25, 26] to be unsatisfactory. Therefore, we propose an improved posterior sampling scheme, named moment matching posterior sampling (MMPS), which we present and motivate in Section 4.2. We evaluate MMPS independently from the context of learning from observations in Appendix E.

**Maximization**  We parameterize our diffusion model $q_\theta(x)$ by a denoiser network $d_\theta(x_t, t)$ and train it via denoising score matching [53, 54], as presented in Section 2. To accelerate the training, we start each iteration with the previous parameters $\theta_k$.

**Initialization**   An important part of our pipeline is the initial prior $q_0(x)$. Any initial prior leads to a local optimum [42–44], but an informed initial prior can reduce the number of iterations until convergence. In our experiments, we take a Gaussian distribution $\mathcal{N}(x \mid \mu_x, \Sigma_x)$ as initial prior and fit its mean and covariance by – you guessed it! – expectation-maximization. Fitting a Gaussian distribution by EM is very fast as the maximization step can be conducted in closed-form, especially for low-rank covariance approximations [67].

An alternative we do not explore in this work would be to use a pre-trained diffusion model as initial prior. Pre-training can be contucted on data we expect to be similar to the latents, such as computer simulations or even video games. The more similar, the faster the EM algorithm converges. However, if the initial prior $q_0(x)$ does not cover latents that are otherwise plausible under the observations, the following priors $q_{\theta_k}(x)$ may not cover these latents either. A conservative initial prior is therefore preferable for scientific applications.

## 4.2   Moment Matching Posterior Sampling

To sample from the posterior distribution $q_\theta(x \mid y) \propto q_\theta(x)\, p(y \mid x)$ of our diffusion prior $q_\theta(x)$, we have to estimate the posterior score $\nabla_{x_t} \log q_\theta(x_t \mid y)$ and plug it into the reverse SDE (3). In this section, we propose and motivate an improved approximation for the posterior score. As this contribution is not bound to the context of EM, we temporarily switch back to the notations of Section 2 where our prior is denoted $p(x)$ instead of $q_\theta(x)$.

**Diffusion posterior sampling**   Thanks to Bayes' rule, the posterior score $\nabla_{x_t} \log p(x_t \mid y)$ can be decomposed into two terms [17, 18, 21–25, 46]

$$\nabla_{x_t} \log p(x_t \mid y) = \nabla_{x_t} \log p(x_t) + \nabla_{x_t} \log p(y \mid x_t). \tag{16}$$

As an estimate of the prior score $\nabla_{x_t} \log p(x_t)$ is already available via the denoiser $d_\theta(x_t, t)$, the remaining task is to estimate the likelihood score $\nabla_{x_t} \log p(y \mid x_t)$. Assuming a differentiable measurement function $\mathcal{A}$ and a Gaussian forward process $p(y \mid x) = \mathcal{N}(y \mid \mathcal{A}(x), \Sigma_y)$, Chung et al. [21] propose the approximation

$$p(y \mid x_t) = \int p(y \mid x)\, p(x \mid x_t)\, \mathrm{d}x \approx \mathcal{N}\left(y \mid \mathcal{A}(\mathbb{E}[x \mid x_t]), \Sigma_y\right) \tag{17}$$

which allows to estimate the likelihood score $\nabla_{x_t} \log p(y \mid x_t)$ without training any other network than $d_\theta(x_t, t) \approx \mathbb{E}[x \mid x_t]$. The motivation behind Eq. (17) is that, when $\sigma_t$ is small, assuming that $p(x \mid x_t)$ is narrowly concentrated around its mean $\mathbb{E}[x \mid x_t]$ is reasonable. However, this approximation is very poor when $\sigma_t$ is not negligible. Consequently, DPS [21] is unstable, does not properly cover the support of the posterior $p(x \mid y)$ and often leads to samples $x$ which are inconsistent with the observation $y$ [22–25].

**Moment matching**   To address these flaws, following studies [22–25] take the covariance $\mathbb{V}[x \mid x_t]$ into account when estimating the likelihood score $\nabla_{x_t} \log p(y \mid x_t)$. Specifically, they consider the Gaussian approximation

$$q(x \mid x_t) = \mathcal{N}\left(x \mid \mathbb{E}[x \mid x_t], \mathbb{V}[x \mid x_t]\right) \tag{18}$$

which is closest to $p(x \mid x_t)$ in Kullback-Leibler (KL) divergence [68]. Then, assuming a linear Gaussian forward process $p(y \mid x) = \mathcal{N}(y \mid Ax, \Sigma_y)$, we obtain [68]

$$q(y \mid x_t) = \int p(y \mid x)\, q(x \mid x_t)\, \mathrm{d}x = \mathcal{N}\left(y \mid A\mathbb{E}[x \mid x_t], \Sigma_y + A\mathbb{V}[x \mid x_t]A^\top\right) \tag{19}$$

which allows to estimate the likelihood score $\nabla_{x_t} \log p(y \mid x_t)$ as

$$\nabla_{x_t} \log q(y \mid x_t) = \nabla_{x_t} \mathbb{E}[x \mid x_t]^\top A^\top \left(\Sigma_y + A\mathbb{V}[x \mid x_t]A^\top\right)^{-1} \left(y - A\mathbb{E}[x \mid x_t]\right) \tag{20}$$

under the assumption that the derivative of $\mathbb{V}[x \mid x_t]$ with respect to $x_t$ is negligible [24, 25]. Even with simple heuristics for $\mathbb{V}[x \mid x_t]$, such as $\Sigma_t$ [20] or $(\Sigma_t^{-1} + \Sigma_x^{-1})^{-1}$ [22, 23], this adaptation leads to significantly more stable sampling and better coverage of the posterior $p(x \mid y)$ than DPS [21]. However, we find that heuristics lead to overly dispersed posteriors $q(x_t \mid y) \propto p(x_t)\, q(y \mid x_t)$ in the presence of local covariances – *i.e.* covariances in the neighborhood of $x_t$.

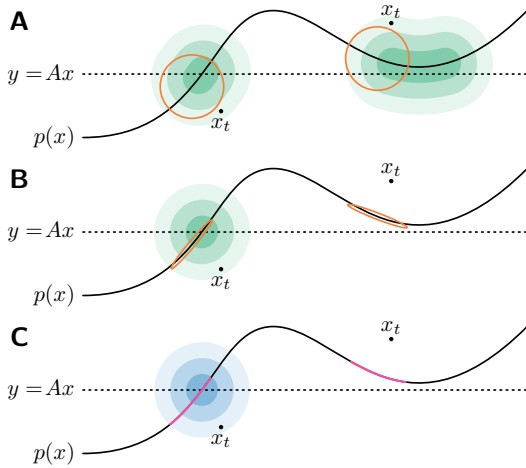

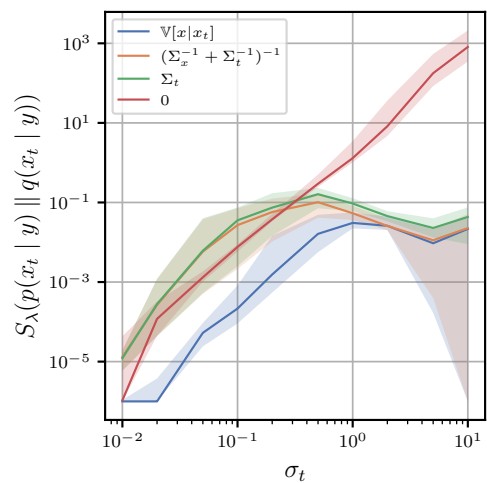

Figure 1. Illustration of the posterior $q(x_t \mid y)$ for the Gaussian approximation $q(x \mid x_t)$ when the prior $p(x)$ lies on a manifold. Ellipses represent 95 % credible regions of $q(x \mid x_t)$. (**A**) With $\Sigma_t$ as heuristic for $\mathbb{V}[x \mid x_t]$, any $x_t$ whose mean $\mathbb{E}[x \mid x_t]$ is close to the plane $y = Ax$ is considered likely. (**B**) With $\mathbb{V}[x \mid x_t]$, more regions are correctly pruned. (**C**) Ground-truth $p(x_t \mid y)$ and $p(x \mid x_t)$ for reference.

Figure 2. Sinkhorn divergence [69] between the posteriors $p(x_t \mid y)$ and $q(x_t \mid y)$ for different heuristics of $\mathbb{V}[x \mid x_t]$ when the prior $p(x)$ lies on 1-d manifolds embedded in $\mathbb{R}^3$. Lines and shades represent the 25-50-75 percentiles for 64 randomly generated manifolds [71] and measurement matrices $A \in \mathbb{R}^{1 \times 3}$. Using $\mathbb{V}[x \mid x_t]$ instead of heuristics leads to orders of magnitude more accurate posteriors $q(x_t \mid y)$.

We illustrate this behavior in Figure 1 and measure its impact as the Sinkhorn divergence [69, 70] between the posteriors $p(x_t \mid y)$ and $q(x_t \mid y)$ when the prior $p(x)$ lies on randomly generated 1-dimensional manifolds [71] embedded in $\mathbb{R}^3$. The prior $p(x)$ is modeled as a mixture of isotropic Gaussians centered around points of the manifold, which gives access to $p(x_t)$, $\mathbb{E}[x \mid x_t]$ and $\mathbb{V}[x \mid x_t]$ analytically. The results, presented in Figure 2, indicate that using $\mathbb{V}[x \mid x_t]$ instead of heuristics leads to orders of magnitude more accurate posteriors $q(x_t \mid y)$. We expect this gap to further increase with real high-dimensional data as the latter often lies along low-dimensional manifolds and, therefore, presents strong local covariances.

Because the MCEM algorithm is sensitive to the accuracy of posterior samples [63–66], we choose to estimate $\mathbb{V}[x \mid x_t]$ using Tweedie's covariance formula [55–58]

$$\mathbb{V}[x \mid x_t] = \Sigma_t + \Sigma_t \nabla^2_{x_t} \log p(x_t) \Sigma_t \tag{21}$$

$$= \Sigma_t \nabla^\top_{x_t} \mathbb{E}[x \mid x_t] \approx \Sigma_t \nabla^\top_{x_t} d_\theta(x_t, t). \tag{22}$$

**Conjugate gradient method** As noted by Finzi et al. [24], explicitly computing and materializing the Jacobian $\nabla^\top_{x_t} d_\theta(x_t, t) \in \mathbb{R}^{N \times N}$ is intractable in high dimension. Furthermore, even if we had access to $\mathbb{V}[x \mid x_t]$, naively computing the inverse of the matrix $\Sigma_y + A\mathbb{V}[x \mid x_t]A^\top$ in Eq. (20) would still be intractable. Fortunately, we observe that the matrix $\Sigma_y + A\mathbb{V}[x \mid x_t]A^\top$ is symmetric positive definite (SPD) and, therefore, compatible with the conjugate gradient (CG) method [72]. The CG method is an iterative algorithm to solve linear systems of the form $Mv = b$ where the SPD matrix $M$ and the vector $b$ are known. Importantly, the CG method only requires implicit access to $M$ through an operator that performs the matrix-vector product $Mv$ given a vector $v$. In our case, the linear system to solve is

$$y - A\mathbb{E}[x \mid x_t] = \left( \Sigma_y + A\mathbb{V}[x \mid x_t]A^\top \right) v \tag{23}$$

$$= \Sigma_y v + A \big( \underbrace{v^\top A \, \Sigma_t \nabla^\top_{x_t} \mathbb{E}[x \mid x_t]}_{\text{vector-Jacobian product}} \big)^\top. \tag{24}$$

Within automatic differentiation frameworks [73, 74], the vector-Jacobian product in the right-hand side can be cheaply evaluated. In practice, due to numerical errors and imperfect training, the Jacobian

$\nabla_{x_t}^\top d_\theta(x_t, t) \approx \nabla_{x_t}^\top \mathbb{E}[x \mid x_t]$ is not always perfectly SPD. Consequently, the CG method becomes unstable after a number of iterations and fails to reach an exact solution. Fortunately, we find that truncating the CG algorithm to very few iterations (1 to 3) already leads to significant improvements over using heuristics for the covariance $\mathbb{V}[x \mid x_t]$. Alternatively, the CG method can be replaced by other iterative algorithms that can solve non-symmetric non-definite linear systems, such as GMRES [75] or BiCGSTAB [76], at the cost of slower convergence.

## 5 Results

We conduct three experiments to demonstrate the effectiveness of our method. We design the first experiment around a low-dimensional latent variable $x$ whose ground-truth distribution $p(x)$ is known. In this setting, we can use asymptotically exact sampling schemes such as predictor-corrector sampling [23, 46] or twisted diffusion sampling [77] without worrying about their computational cost. This allows us to validate our expectation-maximization pipeline (see Algorithm 1) in the limit of (almost) exact posterior sampling. The remaining experiments target two benchmarks from previous studies: corrupted CIFAR-10 and accelerated MRI. These tasks provide a good understanding of how our method would perform in typical empirical Bayes applications with limited data and compute.

### 5.1 Low-dimensional manifold

In this experiment, the latent variable $x \in \mathbb{R}^5 \sim p(x)$ lies on a random 1-dimensional manifold embedded in $\mathbb{R}^5$ represented in Figure 7. Each observation $y \in \mathbb{R}^2 \sim \mathcal{N}(y \mid Ax, \Sigma_y)$ is the result of a random linear projection of a latent $x$ plus isotropic Gaussian noise ($\Sigma_y = 10^{-4}I$). The rows of the measurement matrix $A \in \mathbb{R}^{2 \times 5}$ are drawn uniformly from the unit sphere $\mathbb{S}^4$. We note that observing all push-forward distributions $p(u^\top x)$ with $u \in \mathbb{S}^{N-1}$ of a distribution $p(x)$ in $\mathbb{R}^N$ is sufficient to recover $p(x)$ in theory [78, 79]. In practice, we generate a finite training set of $2^{16}$ pairs $(y, A)$.

We train a DM $q_\theta(x)$ parameterized by a multi-layer perceptron $d_\theta(x_t, t)$ for $K = 32$ EM iterations. We apply Algorithm 3 to estimate the posterior score $\nabla_{x_t} \log q_\theta(x_t \mid y, A)$, but rely on the predictor-corrector [23, 46] sampling scheme with a large number (4096) of correction steps to sample from the posterior $q_\theta(x \mid y, A)$. We provide additional details such as noise schedule, network architectures, and learning rate in Appendix C.

As expected, the model $q_{\theta_k}(x)$ converges towards a stationary distribution whose marginals are close to the marginals of the ground-truth $p(x)$, as illustrated in Figure 3. We attribute the remaining artifacts to finite data and inaccuracies in our sampling scheme.

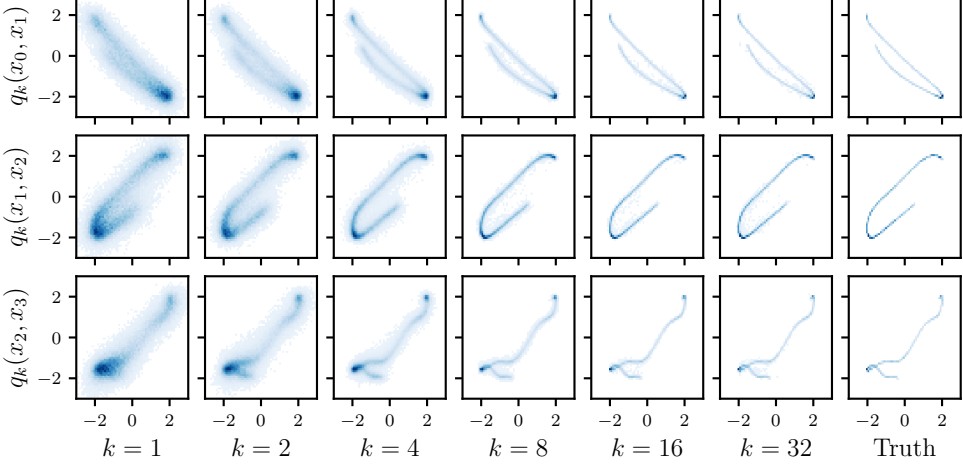

Figure 3. Illustration of 2-d marginals of the model $q_{\theta_k}(x)$ along the EM iterations. The initial Gaussian prior $q_0(x)$ leads to a very dispersed first model $q_{\theta_1}(x)$. The EM algorithm gradually prunes the density regions which are inconsistent with observations, until it reaches a stationary distribution. The marginals of the final distribution are close to the marginals of the ground-truth distribution.

| Method | $\rho$ | FID $\downarrow$ | IS $\uparrow$ |
|---|---|---|---|
| AmbientDiffusion [80] | 0.20 | 11.70 | 7.97 |
| | 0.40 | 18.85 | 7.45 |
| | 0.60 | 28.88 | 6.88 |
| Ours w/ Tweedie | 0.25 | 5.88 | 8.83 |
| | 0.50 | 6.76 | 8.75 |
| | 0.75 | 13.18 | 8.14 |
| Ours w/ $(I + \Sigma_t^{-1})^{-1}$ | 0.75 | 39.94 | 7.69 |
| Ours w/ $\Sigma_t$ | 0.75 | 118.69 | 4.23 |

Table 1. Evaluation of final models trained on corrupted CIFAR-10. Our method outperforms AmbientDiffusion [80] at similar corruption levels. Using heuristics for $\mathbb{V}[x \mid x_t]$ instead of Tweedie's formula greatly decreases the sample quality.

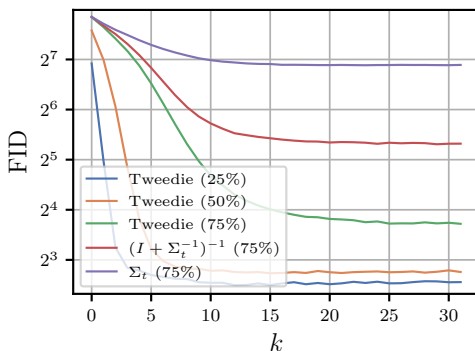

Figure 4. FID of $q_{\theta_k}(x)$ along the EM iterations for the corrupted CIFAR-10 experiment.

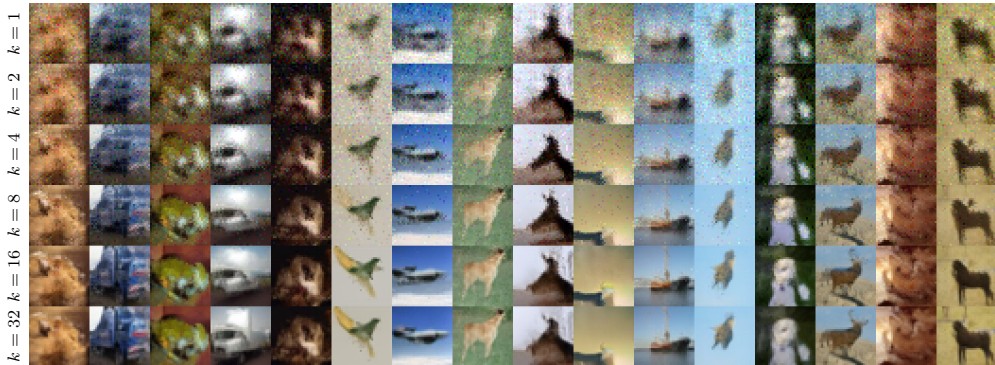

Figure 5. Example of samples from the model $q_{\theta_k}(x)$ along the EM iterations for the corrupted CIFAR-10 experiment with $\rho = 0.75$. We use the deterministic DDIM [50] sampling scheme for easier comparison. Generated images become gradually more detailed and less noisy.

## 5.2 Corrupted CIFAR-10

Following Daras et al. [80], we take the 50 000 training images of the CIFAR-10 [81] dataset as latent variables $x$. A single observation $y$ is generated for each image $x$ by randomly deleting pixels with probability $\rho$. The measurement matrix $A$ is therefore a binary diagonal matrix. We add negligible isotropic Gaussian noise ($\Sigma_y = 10^{-6}I$) for fair comparison with AmbientDiffusion [80] which cannot handle noisy observations.

For each corruption rate $\rho \in \{0.25, 0.5, 0.75\}$, we train a DM $q_\theta(x)$ parameterized by a U-Net [82] inspired network $d_\theta(x_t, t)$ for $K = 32$ EM iterations. We apply Algorithm 2 with $T = 256$ discretization steps and $\eta = 1$ to approximately sample from the posterior $q_\theta(x \mid y, A)$. We apply Algorithm 3 with several heuristics for $\mathbb{V}[x \mid x_t]$ to compare their results against Tweedie's covariance formula. For the latter, we truncate the conjugate gradient method in Algorithm 4 to a single iteration.

For each model $q_{\theta_k}(x)$, we generate a set of 50 000 images and evaluate its Inception score (IS) [83] and Fréchet Inception distance (FID) [84] against the uncorrupted training set of CIFAR-10. We report the results in Table 1 and Figures 4 and 5. At 75 % of corruption, our method performs similarly to AmbientDiffusion [80] at only 20 % of corruption. On the contrary, using heuristics for $\mathbb{V}[x \mid x_t]$ leads to poor sample quality.

## 5.3 Accelerated MRI

Magnetic resonance imaging (MRI) is a non-invasive medical imaging technique used in radiology to inspect the internal anatomy and physiology of the body. MRI measurements of an object are obtained in the frequency domain, also called $k$-space, using strong magnetic fields. However, measuring the

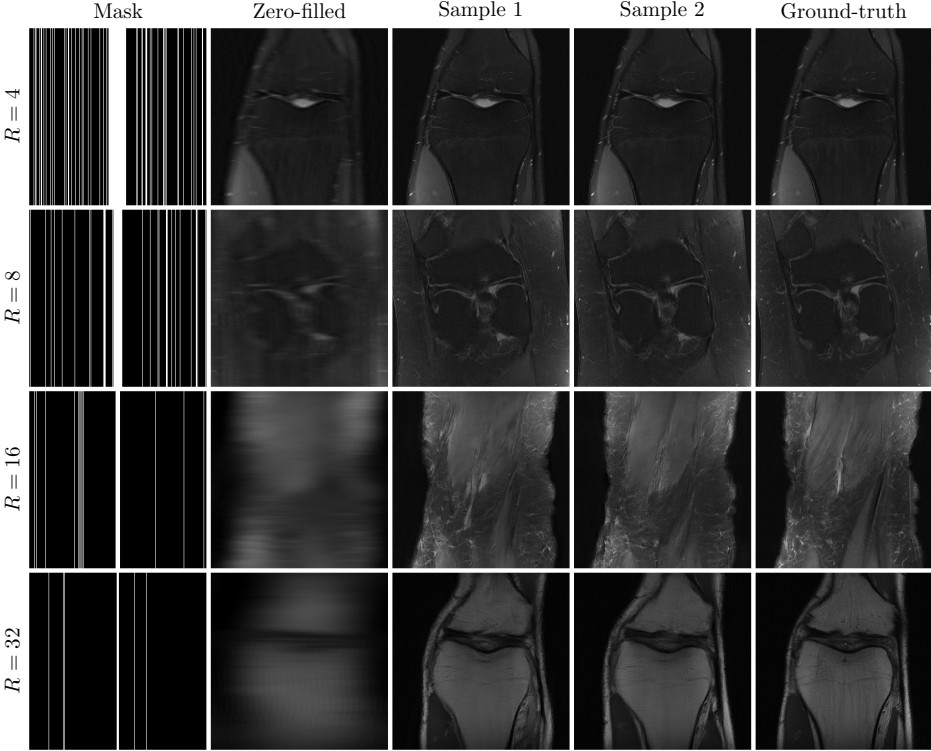

Figure 6. Examples of posterior samples for accelerated MRI using a diffusion prior trained from $k$-space observations only. Posterior samples are detailed and present plausible variations, while remaining consistent with the observation. We provide the zero-filled inverse, where missing frequencies are set to zero, as baseline.

entire $k$-space can be time-consuming and expensive. Accelerated MRI [4–8] consists in inferring the scanned object based on partial, possibly randomized and noisy, frequency measurements.

In this experiment, following Kawar et al. [85], we consider the single-coil knee MRI scans from the fastMRI [7, 8] dataset. We treat each slice between the 10th and 40th of each scan as an independent latent variable $x$, represented as a $320 \times 320$ gray-scale image. Scans are min-max normalized such that pixel values range between $-2$ and $2$. A single observation $y$ is generated for each slice $x$ by first applying the discrete Fourier transform and then a random horizontal frequency sub-sampling with acceleration factor $R = 6$ [85, 86], meaning that a proportion $1/R$ of all frequencies are observed on average. Eventually, we obtain $24\,853$ $k$-space observations to which we add isotropic Gaussian noise ($\Sigma_y = 10^{-4}I$) to match Kawar et al. [85].

Once again, we train a DM $q_\theta(x)$ parameterized by a U-Net [82] inspired network $d_\theta(x_t, t)$ for $K = 16$ EM iterations. We apply Algorithm 2 with $T = 64$ discretization steps and $\eta = 1$ to approximately sample from the posterior $q_\theta(x \mid y, A)$ and truncate the conjugate gradient method in Algorithm 4 to 3 iterations. After training, we employ our final model $q_{\theta_K}(x)$ as a diffusion prior for accelerated MRI. Thanks to our moment matching posterior sampling, we are able to infer plausible scans at acceleration factors up to $R = 32$, as shown in Figure 6. Our samples are noticeably more detailed than the ones of Kawar et al. [85]. We choose not to report the PSNR/SSIM of our samples as these metrics only make sense for regression tasks and unfairly penalize proper generative models [87, 88]. We provide prior samples in Figure 13 and posterior samples for another kind of forward process in Figure 14.

## 6  Related Work

**Empirical Bayes**  A number of previous studies have investigated the use of deep learning to solve the empirical Bayes problem. Louppe et al. [35] use adversarial training for learning a prior

distribution that reproduces the empirical distribution of observations when pushed through a non-differentiable black-box forward process. Dockhorn et al. [33] use normalizing flows [89, 90] to estimate the prior density by variational inference when the forward process consists of additive noise. Vandegar et al. [36] also use normalizing flows but consider black-box forward processes for which the likelihood $p(y \mid x)$ is intractable. They note that empirical Bayes is an ill-posed problem in that distinct prior distributions may result in the same distribution over observations. Vetter et al. [37] address this issue by targeting the prior distribution of maximum entropy while minimizing the sliced-Wasserstein distance [78, 79] with the empirical distribution of observations. These methods rely on generative models $q_\theta(x)$ for which the density $q_\theta(x)$ or samples $x \sim q_\theta(x)$ are differentiable with respect to the parameters $\theta$, which is not or hardly the case for diffusion models.

Closer to this work, Daras et al. [80] and Kawar et al. [85] attempt to train DMs from linear observations only. Daras et al. [80] consider noiseless observations of the form $y = Ax$ and train a network $d_\theta(Ax_t, A, t)$ to approximate $\mathbb{E}[x \mid Ax_t]$ under the assumption that $\mathbb{E}[A^\top A]$ is full-rank. The authors argue that $\mathbb{E}[x \mid Ax_t]$ can act as a surrogate for $\mathbb{E}[x \mid x_t]$. Similarly, Kawar et al. [85] assume Gaussian observations $y \sim \mathcal{N}(y \mid Ax, \Sigma_y)$ and train a network $d_\theta(Px_t, t)$ to approximate $\mathbb{E}[x \mid Px_t]$ under the assumption that $\mathbb{E}[P]$ is full-rank where $P = A^+A$ and $A^+$ is the Moore-Penrose pseudo-inverse of $A$. The authors assume that $d_\theta(Px_t, t)$ can generalize to $P = I$, even if the training data does not contain $P = I$. In both cases, the trained networks are not proper denoisers approximating $\mathbb{E}[x \mid x_t]$ and cannot reliably parameterize a standard diffusion model, which is problematic for downstream applications. Notably, in the case of Bayesian inference, they require custom posterior sampling schemes such as the one proposed by Aali et al. [91] for AmbientDiffusion [80] models. Conversely, in this work, we do not make modifications to the denoising score matching objective [53, 54], which guarantees a proper DM that is compatible with any posterior sampling scheme at every iteration. In addition, we find that our method leads to quantitatively and qualitatively better diffusion priors.

In a concurrent work, Daras et al. [92] propose an algorithm to train DMs from noisy ($A = I$ and $\Sigma_y = \sigma_y^2 I$) data by enforcing the "consistency" of the denoiser along diffusion paths. They prove that the mean $\mathbb{E}[x \mid x_t]$ is the unique consistent denoiser. Interestingly, this training algorithm also relies on posterior samples, which are easy to obtain thanks to the white noise assumption.

**Posterior sampling**   Recently, there has been much work on conditional generation using unconditional diffusion models, most of which adopt the posterior score decomposition in Eq. (16). As covered in Section 4.2, Chung et al. [21] propose an analytical approximation for the likelihood score $\nabla_{x_t} \log p(y \mid x_t)$ when the forward process $p(y \mid x)$ is Gaussian. Song et al. [22] and Rozet et al. [23] improve this approximation by taking the covariance $\mathbb{V}[x \mid x_t]$ into account in the form of simple heuristics. We build upon this idea and replace heuristics with a proper estimate of the covariance $\mathbb{V}[x \mid x_t]$ based on Tweedie's covariance formula [55–58]. Finzi et al. [24] take the same approach, but materialize the matrix $A\mathbb{V}[x \mid x_t]A^\top$ which is intractable in high dimension. Boys et al. [25] replace the covariance $\mathbb{V}[x \mid x_t]$ with a row-sum approximation $\operatorname{diag}(e^\top \mathbb{V}[x \mid x_t])$ where $e$ is the all-ones vector. This approximation is only valid when $\mathbb{V}[x \mid x_t]$ is diagonal, which limits its applicability. Instead, we take advantage of the conjugate gradient method [72] to avoid materializing $A\mathbb{V}[x \mid x_t]A^\top$. A potential cheaper solution is to train a sparse approximation of $\mathbb{V}[x \mid x_t]$, as proposed by Peng et al. [93], but this approach is less general and not immediately applicable to any diffusion model.

A parallel line of work [94–96] extends the conditioning of diffusion models to arbitrary loss terms $\ell(x, y) \propto -\log p(y \mid x)$, for which no reliable analytical approximation of the likelihood score $\nabla_{x_t} \log p(y \mid x_t)$ exists. Song et al. [94] rely on Monte Carlo approximations of the likelihood $p(y \mid x_t) = \int p(y \mid x) \, p(x \mid x_t) \, dx$ by sampling from a Gaussian approximation of $p(x \mid x_t)$. Conversely, He et al. [96] use the mean $\mathbb{E}[x \mid x_t]$ as a point estimate for $p(x \mid x_t)$, but leverage a pre-trained encoder-decoder pair to project the updates of $x_t$ within its manifold. We note that our use of the covariance $\mathbb{V}[x \mid x_t]$ similarly leads to updates tangent to the manifold of $x_t$.

Finally, Wu et al. [77] propose a particle-based posterior sampling scheme that is asymptotically exact in the limit of infinitely many particles as long as the likelihood approximation $q(y \mid x_t)$ – here named the *twisting* function – converges to $p(y \mid x)$ as $t$ approaches 0. Using TDS [77] as part of our expectation-maximization pipeline could lead to better results and/or faster convergence, at the cost of computational resources. In addition, the authors note that the efficiency of TDS [77] depends

on how closely the twisting function approximates the exact likelihood. In this regard, our moment matching Gaussian approximation in Eq. (19) could be a good twisting candidate.

# 7 Discussion

To the best of our knowledge, we are the first to formalize the training of diffusion models from corrupted observations as an empirical Bayes [28–31] problem. In this work, we limit our analysis to linear Gaussian forward processes to take advantage of accurate and efficient high-dimensional posterior sampling schemes. This contrasts with typical empirical Bayes methods which target low-dimensional latent spaces and highly non-linear forward processes [33–37]. In addition, as mentionned in Section 6, these EB methods are not applicable to diffusion models. As such, we choose to benchmark our work against similar methods in the diffusion model literature [80, 85], but stress that a proper comparison with previous empirical Bayes methods would be valuable for both communities. We also note that Monte Carlo EM [59–66] can handle arbitrary forward processes $p(y \mid x)$ as long as one can sample from the posterior $q_\theta(x \mid y)$. Therefore, our method could be adapted to any kind of forward processes in the future. We believe that the works of Dhariwal et al. [97] and Ho et al. [98] on diffusion guidance are good avenues for adapting our method to non-linear, non-differentiable, or even black-box forward processes.

From a computational perspective, the iterative nature of our expectation-maximization method is a drawback compared to previous works [80, 85]. Notably, generating enough samples from the posterior can be expensive, although embarrassingly parallelizable. In addition, even though the architecture and training of the model $q_\theta(x)$ itself are simpler than in previous works [80, 85], the sampling step adds a significant amount of complexity, especially as the convergence of our method is sensitive to the quality of posterior samples. In fact, we find that previous posterior sampling methods [21–23, 25, 26] lead to disappointing results, which motivates us to develop a better one.

As such, moment matching posterior sampling (MMPS) is a byproduct of our work. However, it is not bound to the context of learning from observations and is applicable to any linear inverse problem given a pre-trained diffusion prior. In Appendix E, we evaluate MMPS against previous posterior sampling methods for several linear inverse problems on the FFHQ [99] dataset. We find that MMPS consistently outperforms previous methods, both qualitatively and quantitatively. MMPS is remarkably stable and requires fewer sampling steps to generate qualitative samples, which largely makes up for its slightly higher step cost.

Finally, as mentioned in Section 6, empirical Bayes is an ill-posed problem in that distinct prior distributions may result in the same distribution over observations. In other words, it is generally impossible to identify "the" ground-truth distribution $p(x)$ given an empirical distribution of observations $p(y)$. Instead, for a sufficiently expressive diffusion model, our EM method will eventually converge to a prior $q_\theta(x)$ that is consistent with $p(y)$, but generally different from $p(x)$. Following the maximum entropy principle, as advocated by Vetter et al. [37], is left to future work.

## Acknowledgments and Disclosure of Funding

François Rozet and Gérôme Andry are research fellows of the F.R.S.-FNRS (Belgium) and acknowledge its financial support.

The present research benefited from computational resources made available on Lucia, the Tier-1 supercomputer of the Walloon Region, infrastructure funded by the Walloon Region under the grant n°1910247. The computational resources have been provided by the Consortium des Équipements de Calcul Intensif (CÉCI), funded by the Fonds de la Recherche Scientifique de Belgique (F.R.S.-FNRS) under the grant n°2.5020.11 and by the Walloon Region.

MRI data used in the preparation of this article were obtained from the NYU fastMRI Initiative database [7, 8]. As such, NYU fastMRI investigators provided data but did not participate in analysis or writing of this report. A listing of NYU fastMRI investigators, subject to updates, can be found at https://fastmri.med.nyu.edu/. The primary goal of fastMRI is to test whether machine learning can aid in the reconstruction of medical images.

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

# A  Algorithms

---

**Algorithm 1** Expectation-maximization pipeline

---

1  Choose an initial prior $q_0(x)$
2  Initialize the parameters $\theta$ of the denoiser $d_\theta(x_t, t)$
3  **for** $k = 1$ to $K$ **do**
4      **for** $i = 1$ to $S$ **do**
5          $y_i, A_i \sim p(y, A)$
6          $x_i \sim q_{k-1}(x \mid y_i, A_i)$                                      # Posterior sampling
7      **repeat**
8          $i \sim \mathcal{U}(\{1, \ldots, S\})$
9          $t \sim \mathcal{U}(0, 1)$
10         $z \sim \mathcal{N}(0, I)$
11         $x_t \leftarrow x_i + \sigma_t z$
12         $\ell \leftarrow \lambda_t \big\| d_\theta(x_t, t) - x_i \big\|^2$                     # Denoising score matching
13         $\theta \leftarrow \text{GRADIENTDESCENT}(\theta, \nabla_\theta \ell)$
14     **until convergence**
15     $\theta_k \leftarrow \theta$
16     $q_k := q_{\theta_k}$
17 **return** $\theta_K$

---

**Algorithm 2** DDIM-style posterior sampling

---

1  $x_1 \sim \mathcal{N}(0, \Sigma_1)$
2  **for** $i = T$ to $1$ **do**
3      $s \leftarrow {}^{i-1}\!/_T$
4      $t \leftarrow {}^{i}\!/_T$
5      $\hat{x} \leftarrow x_t + \Sigma_t s_\theta(x_t \mid y, A)$                 # Estimate $\mathbb{E}[x \mid x_t, y, A]$
6      $z \sim \mathcal{N}(0, I)$
7      $x_s \leftarrow \hat{x} + \sigma_s \sqrt{1 - \eta \left(1 - \dfrac{\sigma_s^2}{\sigma_t^2}\right)} \dfrac{x_t - \hat{x}}{\sigma_t} + \sigma_s \sqrt{\eta \left(1 - \dfrac{\sigma_s^2}{\sigma_t^2}\right)} z$
8  **return** $x_0$

---

**Algorithm 3** Moment matching posterior score

---

1   **function** $s_\theta(x_t \mid y, A)$                        # Estimate $\nabla_{x_t} \log q_\theta(x_t \mid y, A)$
2      $\hat{x} \leftarrow d_\theta(x_t, t)$
3      **if** Tweedie **then**
4          $\Sigma_{x \mid x_t} \leftarrow \Sigma_t \nabla_{x_t} d_\theta(x_t, t)^\top$
5      **else**
6          $\Sigma_{x \mid x_t} \leftarrow \Sigma_t$ or $(I + \Sigma_t^{-1})^{-1}$ or $(\Sigma_x^{-1} + \Sigma_t^{-1})^{-1}$
7      $u \leftarrow \left(\Sigma_y + A\Sigma_{x \mid x_t} A^\top\right)^{-1} (y - A\hat{x})$     # Solve with conjugate gradient method
8      $s_{y \mid x} \leftarrow \nabla_{x_t} d_\theta(x_t, t)^\top A^\top u$             # Estimate $\nabla_{x_t} \log q_\theta(y \mid x_t, A)$
9      $s_x \leftarrow \Sigma_t^{-1}(\hat{x} - x_t)$                     # Estimate $\nabla_{x_t} \log q_\theta(x_t)$
10    **return** $s_x + s_{y \mid x}$

---

**Algorithm 4** Conjugate gradient method

---

1  **function** CONJUGATEGRADIENT($A, b, x_0$)
2      $r_0 \leftarrow b - Ax_0$
3      $p_0 \leftarrow r_0$
4      **for** $i = 0$ to $N - 1$ **do**
5          **if** $\|r_i\| \leq \epsilon$ **then**
6              **return** $x_i$
7          $\alpha_i \leftarrow \dfrac{r_i^\top r_i}{p_i^\top A p_i}$
8          $x_{i+1} \leftarrow x_i + \alpha_i p_i$
9          $r_{i+1} \leftarrow r_i - \alpha_i A p_i$
10          $\beta_i \leftarrow \dfrac{r_{i+1}^\top r_{i+1}}{r_i^\top r_i}$
11          $p_{i+1} \leftarrow r_i + \beta_i p_i$
12      **return** $x_N$

---

# B  Tweedie's formulae

**Theorem 1.** *For any distribution $p(x)$ and $p(x_t \mid x) = \mathcal{N}(x_t \mid x, \Sigma_t)$, the first and second moments of the distribution $p(x \mid x_t)$ are linked to the score function $\nabla_{x_t} \log p(x_t)$ through*

$$\mathbb{E}[x \mid x_t] = x_t + \Sigma_t \nabla_{x_t} \log p(x_t) \tag{25}$$

$$\mathbb{V}[x \mid x_t] = \Sigma_t + \Sigma_t \nabla_{x_t}^2 \log p(x_t) \Sigma_t \tag{26}$$

We provide proofs of Theorem 1 for completeness, even though it is a well known result [55–58].

*Proof.*

$$
\begin{aligned}
\nabla_{x_t} \log p(x_t) &= \frac{1}{p(x_t)} \nabla_{x_t} p(x_t) \\
&= \frac{1}{p(x_t)} \int \nabla_{x_t} p(x, x_t) \, \mathrm{d}x \\
&= \frac{1}{p(x_t)} \int p(x, x_t) \nabla_{x_t} \log p(x, x_t) \, \mathrm{d}x \\
&= \int p(x \mid x_t) \nabla_{x_t} \log p(x_t \mid x) \, \mathrm{d}x \\
&= \int p(x \mid x_t) \Sigma_t^{-1}(x - x_t) \, \mathrm{d}x \\
&= \Sigma_t^{-1} \mathbb{E}[x \mid x_t] - \Sigma_t^{-1} x_t \qquad\qquad \square
\end{aligned}
$$

*Proof.*

$$
\begin{aligned}
\nabla_{x_t}^2 \log p(x_t) &= \nabla_{x_t} \nabla_{x_t}^\top \log p(x_t) \\
&= \nabla_{x_t} \mathbb{E}[x \mid x_t]^\top \Sigma_t^{-1} - \Sigma_t^{-1} \\
&= \left( \int \nabla_{x_t} p(x \mid x_t) \, x^\top \, \mathrm{d}x \right) \Sigma_t^{-1} - \Sigma_t^{-1} \\
&= \left( \int p(x \mid x_t) \nabla_{x_t} \log \frac{p(x_t \mid x)}{p(x_t)} \, x^\top \, \mathrm{d}x \right) \Sigma_t^{-1} - \Sigma_t^{-1} \\
&= \left( \int p(x \mid x_t) \Sigma_t^{-1}\big(x - \mathbb{E}[x \mid x_t]\big) \, x^\top \, \mathrm{d}x \right) \Sigma_t^{-1} - \Sigma_t^{-1} \\
&= \Sigma_t^{-1} \Big( \mathbb{E}\left[ xx^\top \mid x_t \right] - \mathbb{E}[x \mid x_t] \mathbb{E}[x \mid x_t]^\top \Big) \Sigma_t^{-1} - \Sigma_t^{-1} \\
&= \Sigma_t^{-1} \mathbb{V}[x \mid x_t] \Sigma_t^{-1} - \Sigma_t^{-1} \qquad\qquad \square
\end{aligned}
$$

## C   Experiment details

All experiments are implemented within the JAX [73] automatic differentiation framework. The code for all experiments is made available at https://github.com/francois-rozet/diffusion-priors.

**Diffusion models**   As mentioned in Section 2, in this work, we adopt the variance exploding SDE [45] and the denoiser parameterization [51]. Following Karras et al. [51], we precondition our denoiser $d_\theta(x_t, t)$ as

$$d_\theta(x_t, t) = \frac{1}{\sigma_t^2 + 1} x_t + \frac{\sigma_t}{\sqrt{\sigma_t^2 + 1}} h_\theta \left( \frac{x_t}{\sqrt{\sigma_t^2 + 1}}, \log \sigma_t \right) \tag{27}$$

where $h_\theta(x, \log \sigma)$ is an arbitrary noise-conditional network. The scalar $\log \sigma$ is embedded as a vector using a sinusoidal positional encoding [100]. In our experiments, we use an exponential noise schedule

$$\sigma_t = \exp \left( (1 - t) \log 10^{-3} + t \log 10^2 \right), \tag{28}$$

loss weights $\lambda_t = \frac{1}{\sigma_t^2} + 1$ and sample $t$ from a Beta distribution $\mathcal{B}(\alpha = 3, \beta = 3)$ during training.

**Low-dimensional manifold**   The noise-conditional network $h_\theta(x, \log \sigma)$ is a multi-layer perceptron with 3 hidden layers of 256 neurons and SiLU [101] activation functions. A layer normalization [102] function is inserted after each activation. The input of the network is the concatenation of $x_t$ and the noise embedding vector. We train the network with Algorithm 1 for $K = 32$ EM iterations. Each iteration consists of $16\,384$ optimization steps of the Adam [103] optimizer. The optimizer and learning rate are re-initialized after each EM iteration. Other hyperparameters are provided in Table 2.

Table 2. Hyperparameters for the low-dimensional manifold experiment.

| | |
|---|---|
| Architecture | MLP |
| Input shape | (5) |
| Hidden features | (256, 256, 256) |
| Activation | SiLU |
| Normalization | LayerNorm |
| Optimizer | Adam |
| Weight decay | 0.0 |
| Scheduler | linear |
| Initial learning rate | $1 \times 10^{-3}$ |
| Final learning rate | $1 \times 10^{-6}$ |
| Gradient norm clipping | 1.0 |
| Batch size | 1024 |
| Steps per EM iteration | $16\,384$ |
| EM iterations | 32 |

We apply Algorithm 3 to estimate the posterior score $\nabla_{x_t} \log p(x_t)$ and truncate Algorithm 4 to 3 iterations. We rely on the predictor-corrector [23, 46] sampling scheme to sample from the posterior $q_\theta(x \mid y, A)$. Following Rozet et al. [23], the predictor is a deterministic DDIM [50] step and the corrector is a Langevin Monte Carlo step. We perform 4096 prediction steps, each followed by 1 correction step. At each EM iteration, we generate a single latent $x$ for each pair $(y, A)$.

We generate smooth random manifolds according to a procedure described by Zenke et al. [71]. We evaluate the Sinkhorn divergences using the POT [70] package with an entropic regularization factor $\lambda = 1e - 3$.

**Corrupted CIFAR-10**   The noise-conditional network $h_\theta(x, \log \sigma)$ is a U-Net [82] with residual blocks [104], SiLU [101] activation functions and layer normalization [102]. Each residual block is modulated with respect to the noise $\sigma_t$ in the style of diffusion transformers [105]. A multi-head self-attention block [100] is inserted after each residual block at the last level of the U-Net. We train

the network with Algorithm 1 for $K = 32$ EM iterations. Each iteration consists of 256 epochs over the training set (50 000 images). To prevent overfitting, images are augmented with horizontal flips and hue shifts. The optimizer is re-initialized after each EM iteration. Other hyperparameters are provided in Table 3.

Table 3. Hyperparameters for the corrupted CIFAR-10 and accelerated MRI experiments.

| Experiment | corrupted CIFAR-10 | accelerated MRI |
|---|---|---|
| Architecture | U-Net | U-Net |
| Input shape | (32, 32, 3) | (80, 80, 16) |
| Residual blocks per level | (5, 5, 5) | (3, 3, 3, 3) |
| Channels per level | (128, 256, 384) | (128, 256, 384, 512) |
| Attention heads per level | (0, 4, 0) | (0, 0, 0, 4) |
| Kernel size | 3 | 3 |
| Activation | SiLU | SiLU |
| Normalization | LayerNorm | LayerNorm |
| Optimizer | Adam | Adam |
| Weight decay | 0.0 | 0.0 |
| Learning rate | $2 \times 10^{-4}$ | $10^{-4}$ |
| Gradient norm cliping | 1.0 | 1.0 |
| EMA decay | 0.9999 | 0.999 |
| Dropout | 0.1 | 0.1 |
| Augmentation | h-flip, hue | h-flip, pad & crop |
| Batch size | 256 | 256 |
| Epochs per EM iteration | 256 | 64 |
| EM iterations | 32 | 16 |

We apply Algorithm 2 with $T = 256$ discretization steps and $\eta = 1$ to sample from the posterior $q_\theta(x \mid y, A)$. We apply Algorithm 3 with several heuristics for $\mathbb{V}[x \mid x_t]$ to compare their results against Tweedie's covariance formula. For the latter, we truncate the conjugate gradient method in Algorithm 4 to a single iteration. At each EM iteration, we generate a single latent $x$ for each pair $(y, A)$. Each EM iteration (including sampling and training) takes around 4 h on 4 A100 (40GB) GPUs.

We evaluate the Inception score (IS) [83] and Fréchet Inception distance (FID) [84] of generated images using the `torch-fidelity` [106] package.

**Accelerated MRI**  The noise-conditional network architecture is the same as for the corrupted CIFAR-10 experiment. The $320 \times 320 \times 1$ tensor $x_t$ is reshaped into a $80 \times 80 \times 16$ tensor using pixel shuffling [107] before entering the network. We train the network with Algorithm 1 for $K = 16$ EM iterations. Each iteration consists of 64 epochs over the training set ($2 \times 24\,853$ images). To prevent overfitting, images are augmented with horizontal flips and random crops. The optimizer is re-initialized after each EM iteration. Other hyperparameters are provided in Table 3.

We apply Algorithm 2 with $T = 64$ discretization steps and $\eta = 1$ to sample from the posterior $q_\theta(x \mid y, A)$. We truncate the conjugate gradient method in Algorithm 4 to 3 iterations. At each EM iteration, we generate 2 latents $x$ for each pair $(y, A)$, which acts as data augmentation. Each EM iteration (including sampling and training) takes around 3 h on 4 A100 (40GB) GPUs.

# D  Additional figures

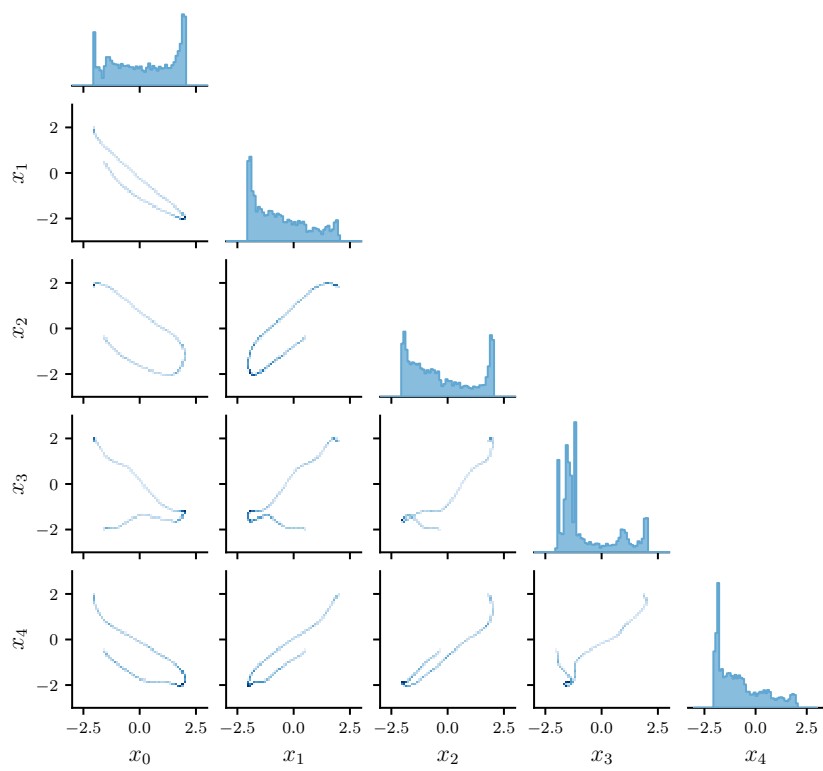

Figure 7. 1-d and 2-d marginals of the ground-truth distribution $p(x)$ used in the low-dimensional manifold experiment. The distribution lies on a random 1-dimensional manifold embedded in $\mathbb{R}^5$.

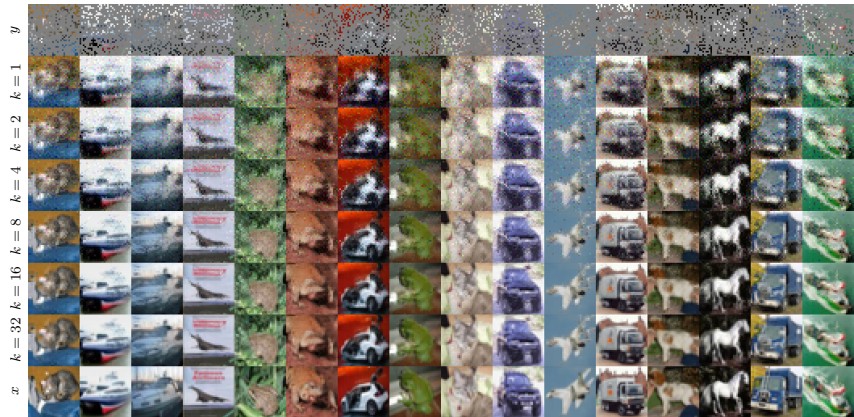

Figure 8. Example of samples from the posterior $q_{\theta_k}(x \mid y)$ along the EM iterations for the CIFAR-10 experiment. The generated images become gradually more detailed and less noisy.

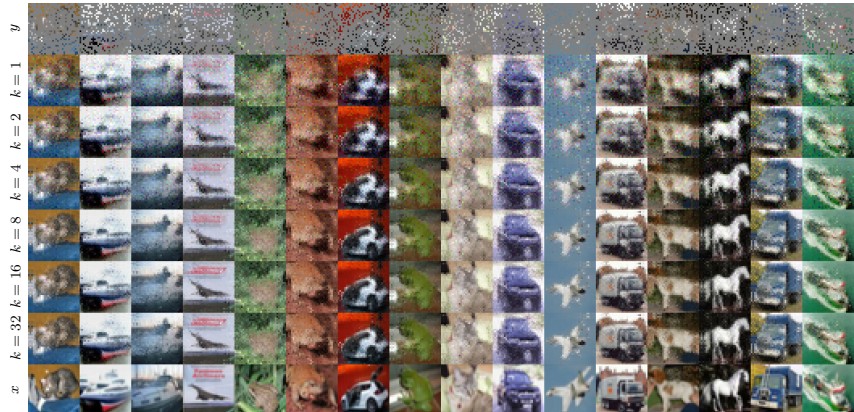

Figure 9. Example of samples from the posterior $q_{\theta_k}(x \mid y)$ along the EM iterations for the CIFAR-10 experiment when the heuristic $(I + \Sigma_t^{-1})^{-1}$ is used for $\mathbb{V}[x \mid x_t]$. The generated images become gradually more detailed but some noise remains.

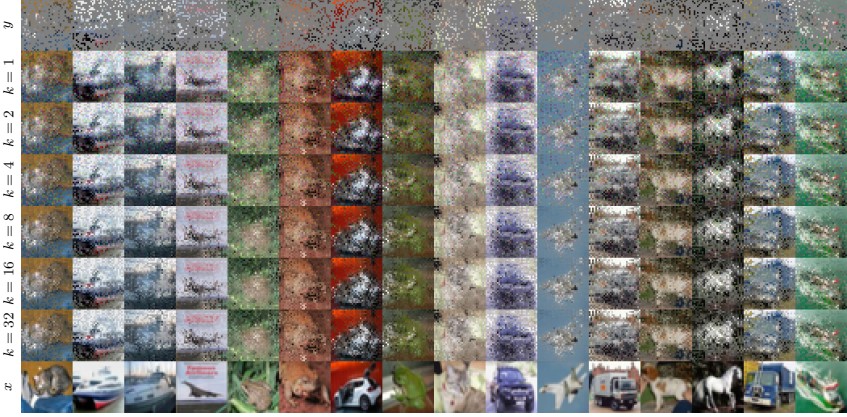

Figure 10. Example of samples from the posterior $q_{\theta_k}(x \mid y)$ along the EM iterations for the CIFAR-10 experiment when the heuristic $\Sigma_t$ is used for $\mathbb{V}[x \mid x_t]$. The generated images remain very noisy.

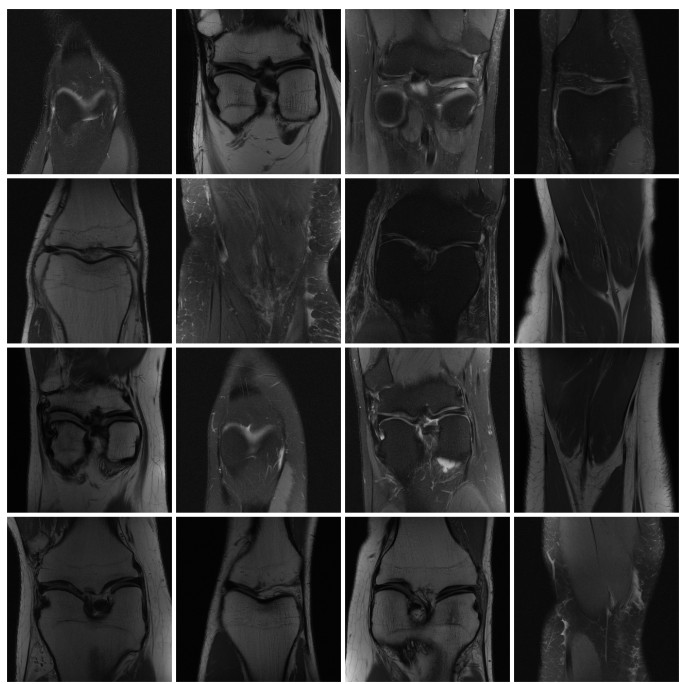

Figure 11. Example of scan slices from the fastMRI [7, 8] dataset.

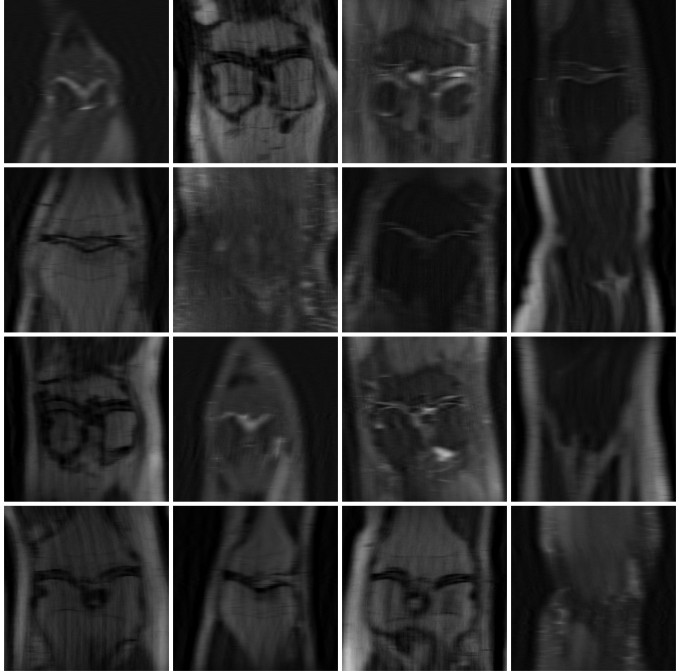

Figure 12. Example of $k$-space sub-sampling observations with acceleration factor $R = 6$ for the accelerated MRI experiment. We represent each observation by its zero-filled inverse, where missing frequencies are set to zero before taking the inverse discrete Fourier transform.

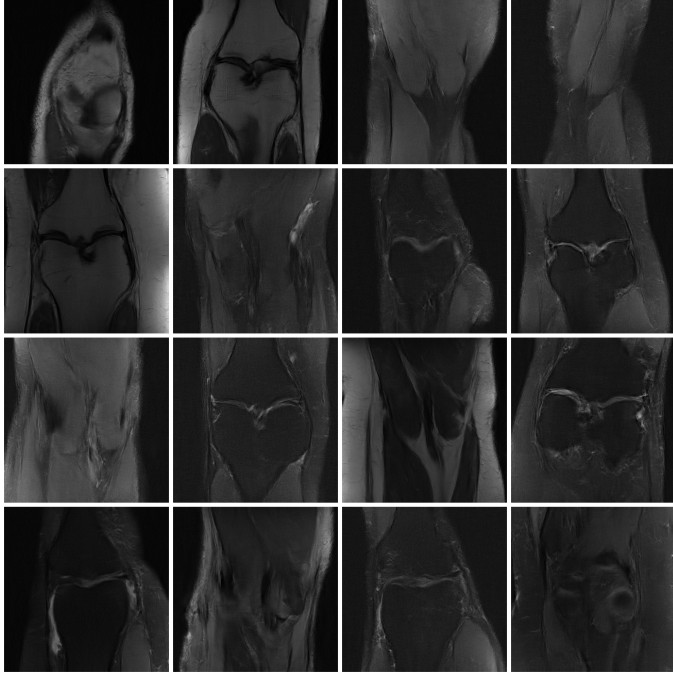

Figure 13. Example of samples from the final model $q_{\theta_k}(x)$ for the accelerated MRI experiment. The samples present varied and coherent global structures. Samples seem slightly less sharp than real scans (see Figure 11), but do not present artifacts typical to unresolved frequencies (see Figure 12).

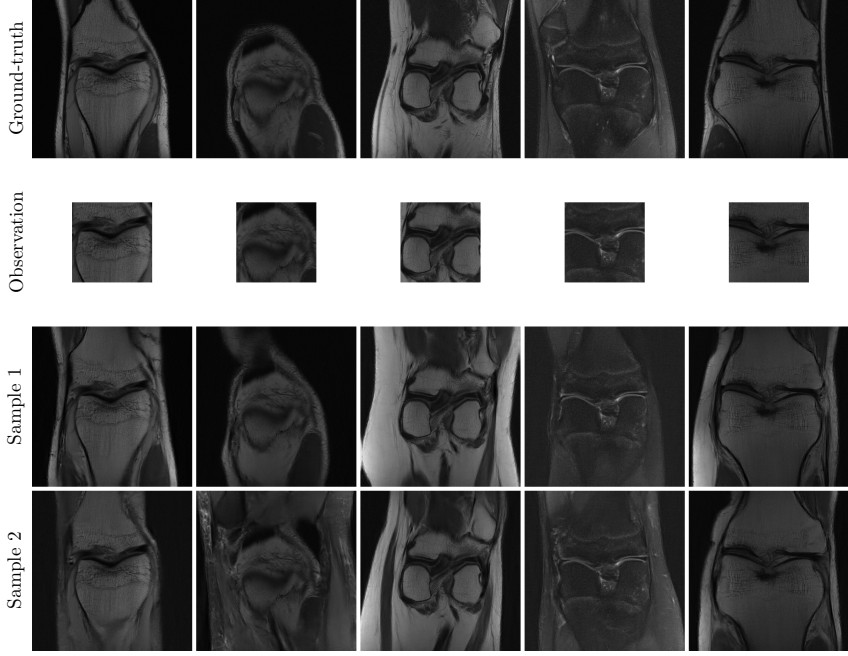

Figure 14. Examples of posterior samples using a diffusion prior trained from $k$-space observations only. The forward process crops the latent $x$ to a centered $160 \times 160$ window. Moment matching posterior sampling is used to sample from the posterior. Samples are consistent with the ground-truth where observed, but present plausible variations elsewhere.

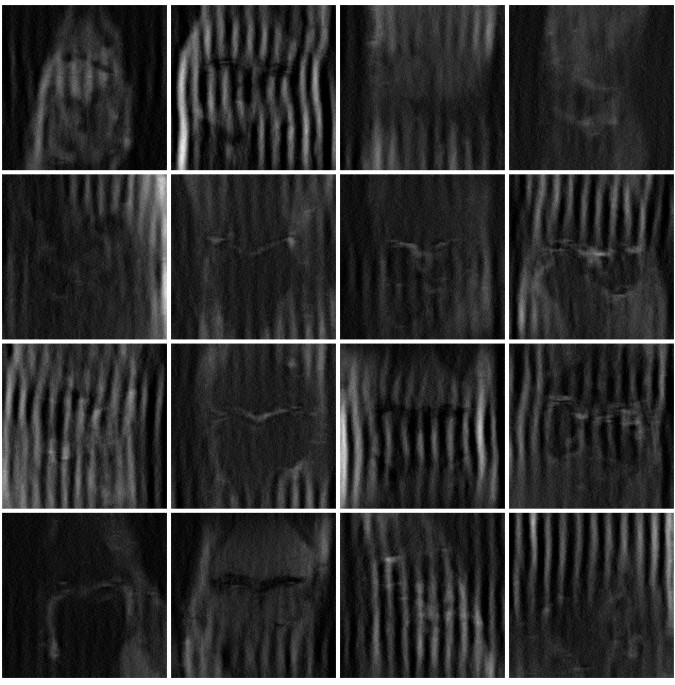

Figure 15. Example of samples from the model $q_{\theta_k}(x)$ after $k = 2$ EM iterations for the accelerated MRI experiment when the heuristic $(I + \Sigma_t^{-1})^{-1}$ is used for $\mathbb{V}[x \mid x_t]$. The samples start to present vertical artifacts due to poor sampling.

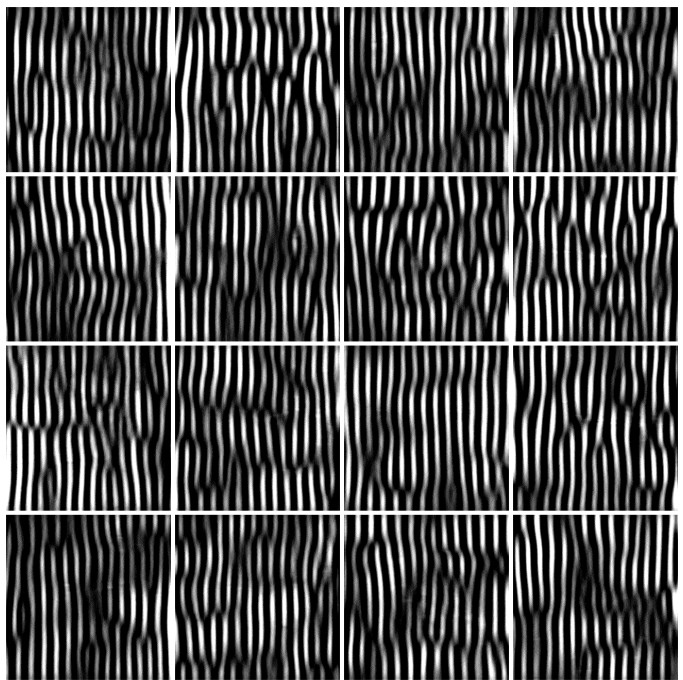

Figure 16. Example of samples from the model $q_{\theta_k}(x)$ after $k = 4$ EM iterations for the accelerated MRI experiment when the heuristic $(I + \Sigma_t^{-1})^{-1}$ is used for $\mathbb{V}[x \mid x_t]$. The artifacts introduced by the poor sampling get amplified at each iteration, leading to a total collapse after few iterations.

# E  Evaluation of MMPS

In this section, we evaluate the moment matching posterior sampling (MMPS) method presented in Section 4.2 independently from the context of learning from observations. The code for this section is made available at https://github.com/francois-rozet/mmps-benchmark.

**Tasks**   We consider four linear inverse problems on the $256 \times 256$ FFHQ [99] dataset. (i) For box inpainting, we mask out a randomly positioned $128 \times 128$ square of pixels and add a large amount of noise ($\sigma_y = 1$). (ii) For random inpainting, we randomly delete pixels with $98\%$ probability and add a small amount of noise ($\sigma_y = 0.01$). (iii) For motion deblur, we apply a randomly generated $61 \times 61$ motion blur kernel and add a medium amount of noise ($\sigma_y = 0.1$). (iv) For super resolution, we apply a $4\times$ bicubic downsampling and add a medium amount of noise ($\sigma_y = 0.1$).

**Methods**   For all inverse problems, we use the pre-trained diffusion model provided by Chung et al. [21] as diffusion prior. We adapt and extend the DPS [21] codebase to support MMPS as well as DiffPIR [26], ΠGDM [22] and TMPD [25]. We use the DDIM [50] sampler with $\eta = 1$ for all methods, which is equivalent to the DDPM [16] sampler. We fine-tune the hyperparameters of DPS ($\zeta' = 0.5$) and DiffPIR ($\lambda = 8.0$) to have the best results across the four tasks. With MMPS, we find that the Jacobian of the pre-trained model provided by Chung et al. [21] is strongly non-symmetric and non-definite for large $\sigma_t$, which leads to unstable conjugate gradient (CG) [72] iterations. We therefore replace the CG solver with the GMRES [75] solver, which can solve non-symmetric non-definite linear systems.

**Protocol**   We generate one observation per inverse problem for 100 images[1] of the FFHQ [99] dataset. We generate a sample for each observation with all considered posterior sampling methods. All methods are executed with the same random seed. We compute three standard image reconstruction metrics – LPIPS [108], PSNR and SSIM [109] – for each sample and report their average in Table 4. We present generated samples for each inverse problem in Figures 17, 18 and 19.

As a side note, we emphasize that reconstruction metrics do not necessarily reflect the accuracy of the inferred posterior distribution, which we eventually care about. For example, PSNR and SSIM [109] favor smooth predictions such as the mean $\mathbb{E}[x \mid y]$ over actual samples from the posterior $p(x \mid y)$. Conversely, LPIPS [108] favors predictions which are *perceptually* similar to the reference, even if they are distorted. In general, it is impossible to simultaneously optimize for all reconstruction metrics [87, 88].

**Results**   MMPS consistently outperforms all baselines, both qualitatively and quantitatively. As expected, performing more solver iterations improves the sample quality, especially when the Gram matrix $AA^\top$ is strongly non-diagonal, which is the case for the motion deblur task. However, the improvement shows rapidly diminishing returns, as the difference between 1 and 3 iterations is much larger than between 3 and 5. MMPS is also remarkably stable with respect to the number of sampling steps in contrast to DPS [21], DiffPIR [26] and ΠGDM [22] which are sensitive to the number of steps and choice of hyperparameters. Finally, MMPS requires fewer sampling steps to reach the same image quality as previous methods, which largely makes up for its slightly higher step cost.

---

[1]Chung et al. [21] do not indicate which subset of FFHQ [99] was used to train their model. Without further information, we choose to use the first 100 images for evaluation, which could lead to biased metrics if the diffusion prior was trained on them. However, since we use the same diffusion prior for all posterior sampling methods, the evaluation remains fair.

Table 4. Quantitative evaluation of MMPS with 1, 3 and 5 solver iterations.

| Method | Steps | Box inpainting | | | Random inpainting | | | Motion deblur | | | Super resolution | | |
|---|---|---|---|---|---|---|---|---|---|---|---|---|---|
| | | LPIPS↓ | PSNR↑ | SSIM↑ | LPIPS↓ | PSNR↑ | SSIM↑ | LPIPS↓ | PSNR↑ | SSIM↑ | LPIPS↓ | PSNR↑ | SSIM↑ |
| DiffPIR [26] | 10 | 0.33 | 19.17 | 0.50 | 0.78 | 10.97 | 0.32 | 0.24 | 24.54 | 0.72 | 0.20 | 26.63 | 0.78 |
| DiffPIR [26] | 100 | 0.30 | 18.15 | 0.54 | 0.68 | 10.26 | 0.25 | 0.19 | 23.97 | 0.70 | 0.17 | 25.24 | 0.73 |
| DiffPIR [26] | 1000 | 0.33 | 17.39 | 0.49 | 0.74 | 9.51 | 0.21 | 0.17 | 23.55 | 0.67 | 0.15 | 24.72 | 0.70 |
| DPS [21] | 10 | 0.64 | 10.41 | 0.34 | 0.58 | 12.68 | 0.43 | 0.75 | 8.63 | 0.27 | 0.58 | 12.01 | 0.41 |
| DPS [21] | 100 | 0.38 | 16.82 | 0.50 | 0.39 | 16.66 | 0.49 | 0.29 | 19.75 | 0.57 | 0.35 | 18.29 | 0.54 |
| DPS [21] | 1000 | 0.22 | 21.01 | 0.64 | 0.19 | 21.90 | 0.66 | 0.18 | 22.91 | 0.66 | 0.16 | 25.02 | 0.72 |
| ΠGDM [22] | 10 | 0.40 | 18.94 | 0.61 | 0.59 | 11.28 | 0.40 | 0.25 | 25.83 | 0.76 | 0.25 | 26.42 | 0.77 |
| ΠGDM [22] | 100 | 0.44 | 18.23 | 0.47 | 0.39 | 17.03 | 0.48 | 0.25 | 22.37 | 0.61 | 0.15 | 25.63 | 0.71 |
| ΠGDM [22] | 1000 | 0.81 | 14.80 | 0.31 | **0.14** | 22.32 | 0.69 | 1.06 | 13.12 | 0.21 | 0.64 | 18.41 | 0.29 |
| TMPD [25] | 10 | 0.36 | 19.90 | 0.64 | 0.59 | 11.08 | 0.40 | 0.27 | 25.28 | 0.74 | 0.26 | 26.07 | 0.76 |
| TMPD [25] | 100 | 0.27 | 19.86 | 0.64 | 0.58 | 10.73 | 0.31 | 0.17 | 26.22 | 0.76 | 0.17 | 26.79 | 0.77 |
| TMPD [25] | 1000 | 0.25 | 19.53 | 0.62 | 0.68 | 9.98 | 0.25 | 0.14 | 25.91 | 0.74 | 0.14 | 26.53 | 0.76 |
| MMPS (1) | 10 | 0.27 | 21.19 | 0.68 | 0.26 | 22.41 | 0.69 | 0.33 | 22.12 | 0.66 | 0.24 | 26.94 | 0.78 |
| MMPS (1) | 100 | 0.20 | 21.19 | 0.67 | 0.18 | 22.18 | 0.69 | 0.20 | 23.92 | 0.71 | 0.15 | 27.32 | 0.79 |
| MMPS (1) | 1000 | **0.19** | 20.77 | 0.64 | 0.18 | 21.94 | 0.66 | 0.16 | 23.83 | 0.69 | 0.12 | 26.92 | 0.77 |
| MMPS (3) | 10 | 0.26 | 21.55 | 0.68 | 0.21 | 23.58 | 0.74 | 0.24 | 25.33 | 0.75 | 0.19 | 27.94 | **0.81** |
| MMPS (3) | 100 | 0.20 | 21.29 | 0.67 | 0.15 | 22.76 | 0.71 | 0.15 | 26.16 | 0.76 | 0.13 | 27.18 | 0.78 |
| MMPS (3) | 1000 | **0.19** | 21.01 | 0.64 | 0.15 | 22.45 | 0.68 | 0.12 | 25.73 | 0.74 | **0.11** | 26.69 | 0.76 |
| MMPS (5) | 10 | 0.23 | **21.73** | **0.69** | 0.20 | **23.72** | **0.75** | 0.20 | **26.70** | **0.78** | 0.18 | **28.02** | **0.81** |
| MMPS (5) | 100 | 0.20 | 21.30 | 0.67 | 0.15 | 22.82 | 0.72 | 0.13 | **26.70** | 0.77 | 0.13 | 27.12 | 0.78 |
| MMPS (5) | 1000 | 0.20 | 20.98 | 0.64 | **0.14** | 22.52 | 0.69 | **0.11** | 26.18 | 0.75 | **0.11** | 26.60 | 0.76 |

Table 5. Time and memory complexity of MMPS for the $4\times$ super resolution task. Each solver iteration increases the time per step by around $16\,\mathrm{ms}$. The maximum memory allocated by MMPS is about $10\,\%$ larger than DPS [21] and ΠGDM [22].

| Method | VJPs | Time [ms/step] | Memory [GB] |
|---|---|---|---|
| DiffPIR [26] | 0 | 30.2 | 0.66 |
| DPS [21] | 1 | 40.5 | 2.29 |
| ΠGDM [22] | 1 | 47.6 | 2.30 |
| TMPD [25] | 2 | 62.2 | 2.52 |
| MMPS (1) | 2 | 58.0 | 2.52 |
| MMPS (3) | 4 | 90.1 | 2.52 |
| MMPS (5) | 6 | 122.1 | 2.52 |

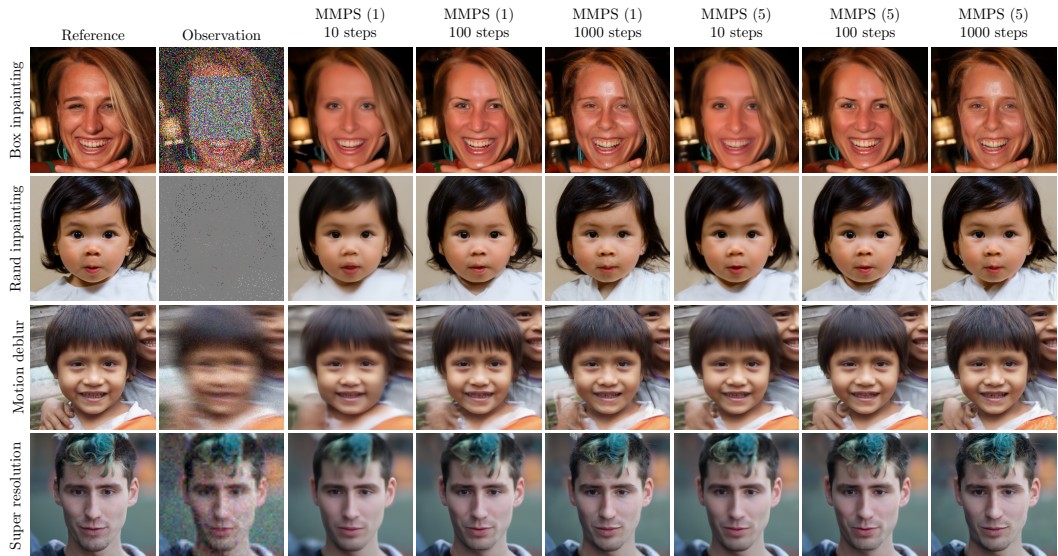

Figure 17. Qualitative evaluation of MMPS with 1 and 5 solver iterations.

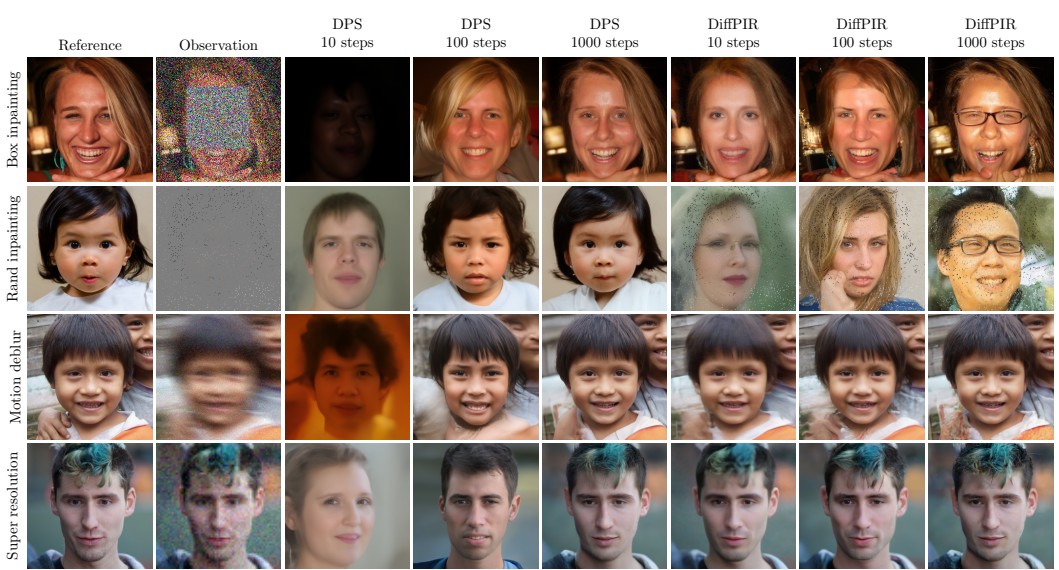

Figure 18. Qualitative evaluation of DPS [21] and DiffPIR [26].

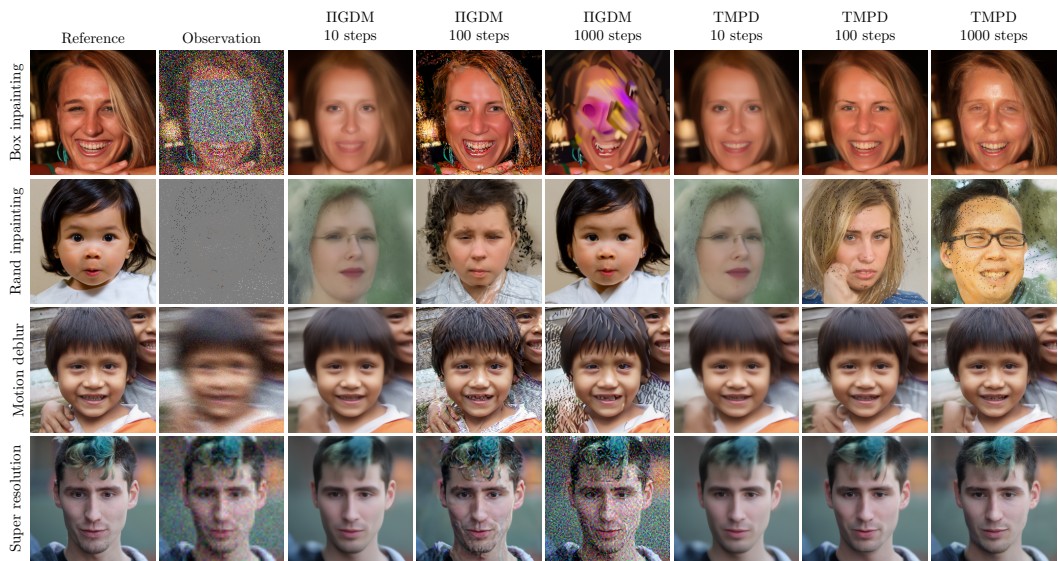

Figure 19. Qualitative evaluation of ΠGDM [22] and TMPD [25].

