# OpenReview forum: "Learning Diffusion Priors from Observations by Expectation Maximization"
_NeurIPS.cc/2024/Conference — NeurIPS 2024 poster_

### Official Review · Reviewer_tnYw · 2024-07-12

**Soundness:** 3
**Presentation:** 3
**Contribution:** 3
**Rating:** 7
**Confidence:** 5

**Summary:**

This paper tackles the problem of learning a diffusion model from incomplete and noisy observations. The problem is modelled as follows; the distribution of the noised and incomplete data is assumed to be $p(y) = \int p(y|x) q(x) dx$. The authors introduce a parametric version of it $p^\theta(y) = \int p(y|x) q^\theta(x) dx$ and then seek to learn $p^\theta$ by minimizing the KL divergence $KL(p||p^\theta)$. As $p^\theta$ is a latent variable model, a natural way to learn it is via the Expectation-Maximization algorithm. The resulting algorithm is iterative and at each step, a diffusion model learned. Also, within each step, the E-step of EM requires sampling from the posterior diffusion $q^\theta(x|y)$ which is intractable in practice. The authors resort to approximate inference and rely on recent advances in posterior sampling of Diffusion models. More specifically, to sample from $q^\theta(x|y)$ it is required to be able to estimate the conditional score within the Diffusion. This conditional score involves computing the gradient log of the conditional distribution $p(y|x_t) = \int p(y|x_0) q^\theta _{0|t}(x_0 | x_t) dx_0$, where the time here referes to the Diffusion steps. The authors propose to replace $q^\theta _{0|t}(x_0 | x_t)$ with its moment projection, which closely follows the work of Boys et al. 2023, and then integrating $p(y|x_0)$ against it. This step however requires an expensive matrix inversion, which the authors propose to approximate via conjugate gradient methods.

**Strengths:**

This paper is well written and presents a natural, original and interesting methodology. The originality stems from the use of the EM algorithm which allows (1) learning at each step a Diffusion model until convergence, (2) using the said Diffusion model for posterior sampling to estimate the expectation. Learning the model until convergence allows using the recent posterior sampling algorithms, which have shown great promise. The experimental setting is also nice and quite diversified; the toy example is welcomed!

**Weaknesses:**

I see a few weaknesses;
- *experiments*: for the posterior sampling algorithm, the authors compare to a narrow set of methods (actually only one method but with different covariance matrices). For example, why is there no comparison with DPS, DDRM or even CoPaint [1], which has been shown to exhibit very good performance (and can easily be extended to noisy inverse problems)? Also, are Figure 1 and 2 obtained using the true covariance matrix? Isn't this a misleading comparison? These figures should be compiled using the proposed method and not the ground-truth covariance. Overall, I think that there isn't enough evidence that the proposed posterior sampler is superior to what is proposed in the literature, given its large memory and time complexity. Especially since the current method requires drawing batches of samples during training, which may hardly fit in a modest GPU due to the computation of the Jacobian.

- *identifiability*: doesn't the proposed framework suffers from an identifiability problem, meaning that the learned model may not actually learn clean data but something else? For example, let's assume that $A$ is the half mask matrix, $x$ is a distribution $q$ over images and $p(y|x,A)$ is a dirac delta at Ax. Then the optimal model can be either $q$ or the law of $A^\dagger X$ with $X \sim q$, where $A^\dagger$ is the pseudo-inverse of $A$. Meaning that the model may as well learn images with missing half. Of course in practice the architecture has inductive biases that help avoid this but surely a UNet cannot simply guess the other half of the images while never seeing these during training. I believe that this is a significant drawback that should be addressed.


[1] Zhang, Guanhua, et al. "Towards coherent image inpainting using denoising diffusion implicit models." (2023).

**Questions:**

N/A

**Limitations:**

see above

---

> ### Author Rebuttal · Authors · 2024-08-05
>
> Thank you for your review and the legitimiate concerns you have raised.
>
> * **W1** (Experiments) We follow your suggestion and benchmark MMPS against previous methods (DPS and $\Pi$GDM). We invite you to consult the global rebuttal regarding these additional experiments.
>
>   Concerning Figures 1 and 2, we use Eq. (20) to compute the covariance $\mathbb{V}[x \mid x_t]$, which is tractable in this toy experiment. We do not see how this would be misleading.
>
>   We also note that we never materialize the Jacobian in the experiments of Section 5. Instead, we leverage the CG method to solve the linear system in Eq. (22), which only requires a cheap vector-Jacobian product per iteration.
>
> * **W2** (Identifiability) Indeed, we forgot to explain how our method would behave when $p(x)$ cannot be uniquely identified from the observations. We propose to replace lines 298-300 with the following paragraph
>
>   > Finally, as mentioned in Section 6, empirical Bayes is an ill-posed problem in that distinct prior distributions may result in the same distribution over observations. In other words, it is generally impossible to identify "the" ground-truth distribution $p(x)$ given an empirical distribution of observations $p(y)$. Instead, for a sufficiently expressive diffusion model, our EM method will eventually converge to a prior $q_\theta(x)$ that is consistent with $p(y)$, but generally different from $p(x)$. In future work, we would like to follow the maximum entropy principle, as advocated by Vetter et al. [36], so as not to reject any possible hypothesis.
>
>   We emphasize that this identifiability issue is a limitation of the problem itself, and not of our method.
>
> We believe that this rebuttal addresses most of your concerns and, therefore, kindly ask you to reconsider your score.

---

> > ### Comment · Reviewer_tnYw · 2024-08-12
> >
> > Thank you for your response and for running the additional experiments.
> >
> > Regarding Figures 1 and 2 I do not see what is their point since you use the true covariance matrix. A more convincing should include the approximation obtained using the Jacobian of the denoiser and then showing that with this approximation one can get decent results, comparable to what you obtain using the true covariance matrix. The current plots tell us nothing about how the approximation< error impacts the final result.
> >
> > This is only a minor weakness however. I leave my score unchanged.

---

> ### Author Response · Authors · 2024-08-12
>
> Thank you for your answer.
>
> > Regarding Figures 1 and 2 I do not see what is their point since you use the true covariance matrix. A more convincing should include the approximation obtained using the Jacobian of the denoiser and then showing that with this approximation one can get decent results, comparable to what you obtain using the true covariance matrix.
>
> Thank you for clarifying your concern here. When the denoiser is trained optimally, that is when $d_\theta(x_t, t) = \mathbb{E}[x \mid x_t]$, there is no approximation in Eq. (21). Tweedie's formula using the Jacobian of the denoiser gives the covariance matrix exactly. Instead of training a denoiser for this toy problem, we assume an optimal denoiser, which ensures that the results are not biased by a choice of parameterization.
>
> > This is only a minor weakness however. I leave my score unchanged.
>
> Given that we have addressed your concerns, we are genuinely surprised by your decision to keep your rating unchanged (weak accept). We would greatly appreciate it if you could describe how to improve our submission.

---

> > ### Comment · Area_Chair_wDgy · 2024-08-13
> >
> > You raised two key weaknesses in your review and the authors addressed both in their rebuttal. Can you provide more detail about how their rebuttal affected your opinion about the paper? The authors asked what more you would have wanted to see to change your score; please respond to this.

---

> > ### Comment · Reviewer_tnYw · 2024-08-13
> >
> > Thank you for your response. I understand that when the denoiser is learned optimally, the covariance is also obtained exactly. My point is still that your approximation relies on taking the jacobian of the denoiser, which is not necessarily a good approximation of the true jacobian of the denoiser (a good parametric $f_\theta$ approximation of a function $f$ does not guarantee that $\nabla_x f_\theta$ is a good parametric approximation of $\nabla_x f$). It would be better to compare with the covariance approximation against the results obtained with the perfect Gaussian approximation.
> >
> > The paper has two contributions; the first one is the EM based algorithm for training a DM with incomplete data and the second contribution is the novel posterior sampler. While the benchmarks for the EM based algorithm are good in my opinion, the ones for the new posterior sampler are rather weak, even though the authors have compared with DPS and PGDM in the rebuttal. Still, this is not enough in my opinion since the comparisons provided are entirely qualitative and are not very convincing. More quantitative benchmarks are required, with comparisons against recent methods that use/do not use the Jacobian of the denoiser. Example: Diffpir [1] or DDNM [2].
> >
> > I have increased my score 7 and I hope that the authors will do their best to strengthen the arguments about their posterior sampler.
> >
> > [1]Zhu, Yuanzhi, Kai Zhang, Jingyun Liang, Jiezhang Cao, Bihan Wen, Radu Timofte, and Luc Van Gool. "Denoising diffusion models for plug-and-play image restoration." In Proceedings of the IEEE/CVF Conference on Computer Vision and Pattern Recognition, pp. 1219-1229. 2023.
> >
> > [2] Wang, Yinhuai, Jiwen Yu, and Jian Zhang. "Zero-shot image restoration using denoising diffusion null-space model." arXiv preprint arXiv:2212.00490 (2022).

---

> > > ### Author Response · Authors · 2024-08-13
> > >
> > > Thank you for taking the time and effort to review our manuscript. Your constructive feedback is deeply appreciated. We will make sure to evaluate MMPS quantitatively and qualitatively against more posterior sampling methods in the camera-ready version.

---

> > > > ### Author Response · Authors · 2024-08-14
> > > >
> > > > Here are preliminary quantitative results. We consider the same linear inverse problems as previously and add two more baselines (TMPD and DiffPIR). We consider 3 standard image quality metrics (LPIPS, PSNR and SSIM) which we average over 100 FFHQ observations. MMPS outperforms all baseline consistently. We note that PSNR and SSIM are biased towards smooth samples while LPIPS favors perceptually detailed samples.
> > > >
> > > > |  |  |  |  | Box inpainting |  |  | Random inpainting |  |  | Motion deblur |  |  | Super resolution |
> > > > |---|---|---|---|---|---|---|---|---|---|---|---|---|---|
> > > > | Method | Steps | LPIPS $\downarrow$ | PSNR $\uparrow$ | SSIM $\uparrow$ | LPIPS $\downarrow$ | PSNR $\uparrow$ | SSIM $\uparrow$ | LPIPS $\downarrow$ | PSNR $\uparrow$ | SSIM $\uparrow$ | LPIPS $\downarrow$ | PSNR $\uparrow$ | SSIM $\uparrow$ |
> > > > | DiffPIR | 10 | 0.97 | 14.23 | 0.13 | 0.78 | 10.97 | 0.32 | 0.39 | 23.98 | 0.51 | 0.31 | 26.45 | 0.69 |
> > > > | DiffPIR | 100 | 0.88 | 15.03 | 0.20 | 0.68 | 10.26 | 0.25 | 0.17 | 25.17 | 0.71 | 0.19 | 26.30 | 0.73 |
> > > > | DiffPIR | 1000 | 0.90 | 14.65 | 0.18 | 0.74 | 9.51 | 0.21 | 0.17 | 24.67 | 0.68 | 0.19 | 25.67 | 0.68 |
> > > > | DPS | 10 | 0.64 | 10.41 | 0.34 | 0.58 | 12.68 | 0.43 | 0.75 | 8.63 | 0.27 | 0.58 | 12.01 | 0.41 |
> > > > | DPS | 100 | 0.38 | 16.82 | 0.50 | 0.39 | 16.66 | 0.49 | 0.29 | 19.75 | 0.57 | 0.35 | 18.29 | 0.54 |
> > > > | DPS | 1000 | 0.22 | 21.02 | 0.64 | 0.19 | 21.90 | 0.66 | 0.18 | 22.91 | 0.66 | 0.16 | 25.02 | 0.72 |
> > > > | PGDM | 10 | 0.48 | 15.32 | 0.57 | 0.52 | 14.44 | 0.47 | 0.46 | 18.61 | 0.54 | 0.37 | 20.47 | 0.67 |
> > > > | PGDM | 100 | 0.92 | 14.18 | 0.22 | 0.23 | 21.59 | 0.66 | 0.84 | 17.02 | 0.34 | 0.45 | 21.80 | 0.51 |
> > > > | PGDM | 1000 | 0.83 | 14.38 | 0.34 | **0.14** | 22.47 | 0.70 | 0.88 | 16.63 | 0.34 | 0.51 | 20.88 | 0.48 |
> > > > | TMPD | 10 | 0.36 | 19.90 | 0.64 | 0.59 | 11.08 | 0.40 | 0.27 | 25.28 | 0.74 | 0.26 | 26.07 | 0.76 |
> > > > | TMPD | 100 | 0.27 | 19.86 | 0.64 | 0.58 | 10.73 | 0.31 | 0.17 | _26.22_ | 0.76 | 0.17 | 26.79 | 0.77 |
> > > > | TMPD | 1000 | 0.25 | 19.53 | 0.62 | 0.68 | 9.98 | 0.25 | 0.14 | 25.91 | 0.74 | 0.14 | 26.53 | 0.76 |
> > > > | MMPS (1) | 10 | 0.27 | 21.19 | _0.68_ | 0.26 | 22.41 | 0.69 | 0.33 | 22.12 | 0.66 | 0.24 | 26.94 | 0.78 |
> > > > | MMPS (1) | 100 | _0.20_ | 21.19 | 0.67 | 0.18 | 22.18 | 0.69 | 0.20 | 23.92 | 0.71 | 0.15 | _27.32_ | _0.79_ |
> > > > | MMPS (1) | 1000 | **0.19** | 20.77 | 0.64 | 0.18 | 21.94 | 0.66 | 0.16 | 23.83 | 0.69 | _0.12_ | 26.92 | 0.77 |
> > > > | MMPS (5) | 10 | 0.23 | **21.73** | **0.69** | 0.20 | **23.72** | **0.75** | 0.20 | **26.70** | **0.78** | 0.18 | **28.02** | **0.81** |
> > > > | MMPS (5) | 100 | _0.20_ | _21.30_ | 0.67 | _0.15_ | _22.82_ | _0.72_ | _0.13_ | **26.70** | _0.77_ | 0.13 | 27.12 | 0.78 |
> > > > | MMPS (5) | 1000 | _0.20_ | 20.98 | 0.64 | **0.14** | 22.52 | 0.69 | **0.11** | 26.18 | 0.75 | **0.11** | 26.60 | 0.76 |

---

### Official Review · Reviewer_mKff · 2024-07-12

**Soundness:** 3
**Presentation:** 3
**Contribution:** 3
**Rating:** 5
**Confidence:** 4

**Summary:**

This paper focuses on training diffusion models using incomplete or noisy data only, which is obtained through a linear measurement operator $A$. While prior works assume full rank of $E[A^TA]$ or $E[A^+A]$ and change the denoising score matching objective while training diffusion models, this paper uses a moment matching posterior sampling approach that does not modify the training objective. Yet, the authors demonstrate superior results both quantitatively and qualitatively on (i) a toy dataset (ii) corrupted CIFAR10 and (iii) accelerated MRI.

**Strengths:**

1. One major strength of the proposed method is that unlike prior works it does not modify the denoising score matching objective, which guarantees a proper diffusion model at every iteration.
2. The experimental results in the toy setting clearly depicts the advantages of using higher order moments in pruning inconsistent regions.
3. The paper is nicely written and the main contributions are clearly stated.

**Weaknesses:**

1. The core idea of moment matching posterior sampling (Section 4.2) has previously appeared in prior works STSL [1] and TMPD [25], where its benefits are clearly demonstrated in large-scale applications.
2. The approximation used in Equation (21) computes gradient of the first order score which is already an approximation. This results in high time and memory complexity, which is precisely the reason why prior works [1,2,25, 65] seek alternatives.
3. What is the typical range of $\sigma_t$ in Fig. 2? It seems like the results are comparable in low noise regime. How does this observation translate into variance preserving SDE, which is the most commonly used form of SDEs for posterior sampling?
4. Does Equation (22) hold for any vector $v$ as the equation suggests?
5. Section 5: Taking gradients of the score becomes an issue especially in high dimensional setting. It is important to understand the time and memory complexity of the proposed algorithm in commonly used benchmarks such as FFHQ or ImageNet ( 256x256 or 512x512).
6. Missing discussion and comparison with other baselines (e.g. [82]) highlighted in the related works Section 6.
7. Theorem 1 in Appendix B is a well known result [63]. It'd be better to cite the original work and provide the key steps only for completeness. Or, the authors should highlight the key distinctions from the prior work.

Missing Related Works

[1] [Beyond First-Order Tweedie: Solving Inverse Problems using Latent Diffusion](https://openaccess.thecvf.com/content/CVPR2024/papers/Rout_Beyond_First-Order_Tweedie_Solving_Inverse_Problems_using_Latent_Diffusion_CVPR_2024_paper.pdf)

[2] [Improving Diffusion Models for Inverse Problems
Using Optimal Posterior Covariance](https://arxiv.org/pdf/2402.02149)

**Questions:**

Please refer to the weaknesses above.

**Limitations:**

Yes, the authors have adequately addressed the limitations.

---

> ### Author Rebuttal · Authors · 2024-08-05
>
> Thank you for your review and the legitimate concerns you have raised.
>
> * **W1** Indeed, we are not the first to use the covariance $\mathbb{V}[x \mid x_t]$ to improve the approximation of the likelihood score. We believe that Finzi et al. [24] were the first to propose it, shortly followed by Boys et al. [25]. As explained in section 6, the difference of our approach resides in the use of the entire covariance matrix, without the need to materialize it (which is intractable) thanks to the conjugate gradient method. Instead, Boys et al. [25] use a row-sum approximation of the covariance's diagonal, which voids some of the benefits of using the covariance as illustrated in Figure 1 an 2. In addition, when the covariance is not diagonal, the row-sum approximation may result in null or negative values, which leads to total failures (NaNs) in our experiments.
>
>   We thank you for bringing STSL [1] to our attention. Are we correct in understanding that STSL uses the trace of the covariance/Hessian to guide the sampling, but does not justify it with a Gaussian approximation of $p(x \mid x_t)$? If so, we are not sure how STSL is similar to the idea of TMPD [25] and/or MMPS.
>
>   In the manuscript we refer a few times to MMPS as a "new" posterior sampling scheme. We propose to replace these occurences with the word "improved". Would this be satisfactory?
>
> * **W2** Materializing the Jacobian $\nabla_{x_t}^\top d_\theta(x_t, t)$ would indeed result in high time and memory complexity, as mentioned at lines 142-144. However, we never do. Instead, we leverage the CG method to solve the linear system in Eq. (22), which only requires a cheap vector-Jacobian product per iteration.
>
> * **W3** In our experiments, $\sigma_t$ ranges between $10^{-3}$ and $10^2$. This figure can be easily translated to the VP SDE with the relation $\bar{\alpha}_t = \frac{1}{1 + \sigma_t^2}$. The divergence should also be scaled accordingly. We note that using the covariance $\mathbb{V}[x | x_t]$ is orders of magnitude better than heuristics, especially in low noise regimes.
>
> * **W4** Yes, this equation simply rewrites $v = (\Sigma_y + A \mathbb{V}[x \mid x_t] A^\top)^{-1} (y - A \mathbb{E}[x \mid x_t])$ and has a solution as long as $\Sigma_y + A \mathbb{V}[x \mid x_t] A^\top$ is invertible, which is the case if $\Sigma_y$ is SPD.
>
> * **W5** We follow your suggestion and benchmark MMPS against previous methods. We invite you to consult the global rebuttal regarding these additional experiments.
>
> * **W6** This is not true. Section 5.3 is dedicated to a comparison with GSURE-Diffusion [82] on the accelerated MRI experiment. We note that Kawar et al. [82] do not provide the identifiers of the scans they use for evaluation in their manuscript or code.
>
> * **W7** Indeed we provide these proofs only for completeness. We propose to add the following sentence in Appendix B for clarity
>
>   > We provide proofs of Theorem 1 for completeness, even though it is a well known result [62-65].
>
> We believe that this rebuttal addresses most of your concerns and, therefore, kindly ask you to reconsider your score.

---

### Official Review · Reviewer_s8XN · 2024-07-12

**Soundness:** 2
**Presentation:** 2
**Contribution:** 2
**Rating:** 4
**Confidence:** 5

**Summary:**

This paper proposes a method to learn generative models from noisy and incomplete data. As opposed to general recent methodologies which require a clean unconditional dataset to solve inverse problems, the paper aims at providing a method to learn the prior from noisy data.

**Strengths:**

The paper is on an interesting and timely topic; this is a much needed class of methods.

**Weaknesses:**

The paper is poorly written, the method is not clear, derivations are incomplete. See my detailed review in the questions part.

**Questions:**

The paper is on an interesting topic as I mentioned however many things are unclear and the algorithm/pipeline needs to be written in a much more clear way before this can be published.

1) The algorithm does not appear at all in the main body of the paper. Please provide a clear and step-by-step demonstration of the training method in the main body of the paper.

2) The authors put a prior on the matrix $A$. What is this prior?

3) It seems that the main difference between the EM and this method is the parameterization of the prior (which is via a score network) described briefly in 4.1. The authors then briefly talk about how to estimate this parameter but then somehow the parameter disappears in the later sections. For example, in eq. (15), the authors say it is easy to estimate the prior score $\nabla_x \log p_t(x)$ but this would require unconditional (and clean!) samples. It is not clear how it is suggested that this quantity is easy to obtain?

4) Section 4.2 describes the standard inverse problem solvers (or a very similar idea). However, as I said in the point above, this requires a pre-trained, clean sampler. This way of writing it is very unclear for readers.

5) Algorithm 1 is the intended "full pipeline" yet the steps are not described clearly. I suggest authors to both move this into the main text as well as extend the discussion of the steps of this pipeline. Some questions are below.

6) In Algorithm 1, is the posterior sampling done by a pretrained diffusion model? If so, this invalidates the main point of the paper. If not, then it means that $q_k$ using the score vector $d_{\theta_k}$, is that a correct conclusion?

7) In Algorithm 1, after sampling $x_i$ for $i = 1,\ldots,S$, does one train the score network with this? In what sense is this an EM algorithm if score matching is done rather than maximum likelihood?

I strongly suggest authors to clarify the pipeline clearly and precisely.

**Limitations:**

Yes

---

> ### Author Rebuttal · Authors · 2024-08-05
>
> We sincerely apologize for the inconvenience caused by the writing of our manuscript. We believe that most of your questions stem from the lack of clarity of Section 4.2, where we fail to state that MMPS is not bound to the EM context and can be applied to any diffusion prior. We propose to clarify this matter by replacing lines 108-112 in Section 4.2 with the following
>
> > To sample from the posterior distribution $q_\theta(x \mid y) \propto q_\theta(x) \, p(y \mid x)$ of our diffusion prior $q_\theta(x)$, we have to estimate the posterior score $\nabla_{x_t} \log q_\theta(x_t \mid y)$ and plug it into the reverse SDE. In this section, we propose and motivate a new approximation for the posterior score. As this contribution is not bound to the context of EM, we temporarily switch back to the notations of Section 2 where our prior is denoted $p(x)$ instead of $q_\theta(x)$.
> >
> > **Diffusion posterior sampling** Thanks to Bayes' rule, the posterior score $\nabla_{x_t} \log p(x_t \mid y)$ can be decomposed into two terms [17, 18, 21-25, 53]
> >
> >  $$ \nabla_{x_t} \log p(x_t \mid y) = \nabla_{x_t} \log p(x_t) + \nabla_{x_t} \log p(y \mid x_t) \, . $$
> >
> > As an estimate of the prior score $\nabla_{x_t} \log p(x_t)$ is already available via the denoiser $d_\theta(x_t, t)$, the remaining task is to estimate the likelihood score $\nabla_{x_t} \log p(y \mid x_t)$.
>
> We now answer your questions, in light of this clarification.
>
> * **Q1** We describe the pipeline in plain text in Section 4.1 and as an algorithm in Appendix A.
>
> * **Q2** The measurement matrix $A$ is defined by the specific instruments that are used to gather the observations. If the configuration or environment of the instruments changes, the measurement matrix $A$ may also change. The prior $p(A)$ is therefore the empirical distribution of $A$ for the observations $y$ we have access to. To paraphrase, we do not specify $p(y, A)$, it is imposed upon us by the task at hand.
>
>   We propose to clarify this matter in the manuscript with the following sentence at line 82
>
>   > For example, if the position or environment of a sensor changes, the measurement matrix $A$ may also change, which leads to an empirical distribution of pairs $(y, A) \sim p(y, A)$.
>
> * **Q3** You are right. The main difference is the parameterization of the prior with a diffusion model. At each step we use the current diffusion prior $q_\theta(x)$ to generate samples from the posterior(s) $q_\theta(x \mid y)$ via the proposed MMPS method.
>
> * **Q4** Indeed, MMPS can be used to solve any linear inverse problem with a diffusion prior. However, this prior does not need to be the distribution of clean data. It can be any prior, including our intermediate priors $q_{\theta_k}(x)$.
>
> * **Q5** We describe the pipeline in plain text in Section 4.1. We unfortunately do not have the space to move Algorithm 1 into the main text.
>
> * **Q6** We do not use a pre-trained diffusion model. As you correctly concluded, $q_k(x) := q_{\theta_k}(x)$ is the current diffusion prior parameterized by the current denoiser $d_{\theta_k}(x_t, t)$.
>
> * **Q7** We show in Section 4 that in the context of EB, the EM step is equivalent to minimizing the KL bewteen $\pi_k(x)$ and $q_{\theta_{k+1}}(x)$. For diffusion models, this KL minimization is generally conducted by denoising score matching or equivalent formulations (e.g. ELBO) using samples from $\pi_k(x)$.
>
> We believe that this rebuttal addresses most of your concerns and, therefore, kindly ask you to reconsider your score.

---

> > ### Comment · Area_Chair_wDgy · 2024-08-13
> >
> > The authors believe their rebuttal addresses most of your concerns; is that true? If not, please say what aspects of their rebuttal are insufficient.

---

### Official Review · Reviewer_XeV5 · 2024-07-13

**Soundness:** 3
**Presentation:** 2
**Contribution:** 3
**Rating:** 5
**Confidence:** 4

**Summary:**

This paper proposes a novel solution to a specific class of Empirical Bayes problems. The method is especially useful when the latent variable and the observations are closely related, such as in cases where the observations are incomplete or noisy latent variables. The major technical obstacle is modeling the posterior distribution for sampling and training, which is addressed through several approximations under Gaussianity.

**Strengths:**

1. **Originality-Middle:** A novel combination of the Bayesian Inverse Problem and Diffusion Model under Gaussianity assumptions.
2. **Quality-Middle:** Well-organized overall, but lacking sufficient details.
3. **Clarity-Middle:** Clear intuitively and qualitatively, but requires more quantitative analysis.
4. **Significance-Middle:** Impressive idea and experimental results, though applications are restricted.

**Weaknesses:**

1. There are not sufficient benchmark experiments, especially those involving other Empirical Bayes methods. A toy example with 3-4 benchmarks, focusing specifically on the trade-off between time consumption and model performance, would help justify the soundness of the model.
2. The validity of the approximation needs to be shown theoretically. See the "Questions" section below.

**Questions:**

1. To what extent does Tweedie's formula violate the assumption that $\mathbb{V}(x|x_t)$ is independent of $x_t$? Are there any bounds or asymptotic results?
2. What is the role of $\Sigma_y$? To what extent does it influence overall performance? What if $\Sigma_y$ is anisotropic or non-diagonal? What if the Gaussian noise is not as small as it is in the experiments?
3. How does the linearity assumption restrict performance? How are the priors over $A$ specified? A real-life example with explicit expressions of $A$ and $\Sigma_y$ would be helpful.
4. Are there any experiments on the improvement per iteration of the Conjugate Gradient method?

**Limitations:**

Yes, the authors adequately addressed the limitations.

---

> ### Author Rebuttal · Authors · 2024-08-05
>
> Thank you for your review and the pertinent questions you have asked.
>
> * **W1** Although we agree that a benchmark with previous empirical Bayes methods would be valuable for the statistical inference community, we don't think it would be relevant to justify our work. First, our method is based on the established EM algorithm, which has stood the test of time. Second, our goal is to train diffusion models from observations as they are best-in-class for modeling high-dimensional distributions and proved to be remarkable priors for Bayesian inference. However, as explained in Section 1 and 6, previous empirical Bayes methods are not applicable to diffusion models. These methods are also typically bound to low-dimensional latent spaces. For these reasons, it is challenging to design a benchmark between our work and these previous EB methods that would be both fair and informative. Instead, we choose to benchmark our work against similar methods in the diffusion model literature.
>
> * **Q1** This assumption is actually the same as the Gaussian approximation of Eq. (17). Indeed, assuming that $p(x \mid x_t)$ is Gaussian, we have (Bishop [67])
>
>   $$ \mathbb{E}[x \mid x_t, y] = \mathbb{E}[x \mid x_t] + \mathbb{V}[x \mid x_t] A^\top (\Sigma_y + A \mathbb{V}[x \mid x_t] A^\top)^{-1} (y - A \mathbb{E}[x \mid x_t]) $$
>
>   but Tweedie's formulae also gives
>
>   $$ \mathbb{E}[x \mid x_t, y] = x_t + \Sigma_t \nabla_{x_t} \log p(x_t \mid y) = \mathbb{E}[x \mid x_t] + \Sigma_t \nabla_{x_t} \log p(y \mid x_t) $$
>
>   Therefore, we have
>
>   $$ \Sigma_t \nabla_{x_t} \log p(y \mid x_t) = \mathbb{V}[x \mid x_t] A^\top (\Sigma_y + A \mathbb{V}[x \mid x_t] A^\top)^{-1} (y - A \mathbb{E}[x \mid x_t]) $$
>
>   which is equivalent to Eq. (20) since $\mathbb{V}[x \mid x_t] = \Sigma_t \nabla_{x_t} \mathbb{E}[x \mid x_t]^\top$.
>
>   Finzi et al. [24] provide a detailed analysis of the Gaussian approximation of Eq. (17) and prove (Theorem 2) that moments of order $n > 2$ converge to 0 at a rate of $\sigma_t^n$.
>
> * **Q2** Our method does not make any assumptions on the covariance $\Sigma_y$. It can be anisotropic and/or non-diagonal. In fact, it is always possible to rewrite the likelihood $\mathcal{N}(y \mid Ax, \Sigma_y)$ as $\mathcal{N}(\Sigma_y^{-1/2} y \mid \Sigma_y^{-1/2} A x, I)$. However, you are right that the signal-to-noise ratio has an impact on our method. We expect larger noise levels to slow down the convergence of the EM algorithm, but lead to an equivalent stationary distribution, under the assumption of infinite data. The rationale is that it is possible to identify all the moments of $p(Ax)$ given $p(y)$ regardless of $\Sigma_y$.
>
> * **Q3** The assumption of a linear Gaussian forward process $p(y \mid x)$ does not "restrict the performance" of our method but limits its applicability. However, many real-life and scientific problems can be formalized with linear Gaussian forward processes. The accelerated MRI experiment is a good example. The measurement matrix $A$ and covariance $\Sigma_y$ are defined by the specific instruments that are used to gather MRI scans. If the configuration or environment of the instruments changes, the measurement matrix $A$ and covariance $\Sigma_y$ may also change. The prior $p(A)$ is therefore the empirical distribution of $A$ for the observations $y$ we have access to. To paraphrase, we do not specify $p(y, A)$, it is imposed upon us by the task at hand.
>
>   We propose to clarify this matter in the manuscript with the following sentence at line 82
>
>   > For example, if the position or environment of a sensor changes, the measurement matrix $A$ may also change, which leads to an empirical distribution of pairs $(y, A) \sim p(y, A)$.
>
> * **Q4** There are no experiments on the improvement per iteration of the CG method currently, but we have conducted additional experiments to benchmark MMPS for this rebuttal and find that increasing the number of CG iterations improves image quality/sharpness, but only marginally and with rapidly diminishing returns. We invite you to consult the global rebuttal regarding these additional experiments.
>
> We believe that this rebuttal addresses most of your concerns and, therefore, kindly ask you to reconsider your score.

---

> > ### Comment · Area_Chair_wDgy · 2024-08-13
> >
> > The authors have provided a detailed response to your questions. How has it affected your opinion about the paper? The authors believe their rebuttal addresses most of your concerns; is that true?

---

### Official Review · Reviewer_3yec · 2024-07-15

**Soundness:** 3
**Presentation:** 4
**Contribution:** 3
**Rating:** 8
**Confidence:** 5

**Summary:**

The authors propose a new framework for training diffusion models from corrupted data, based on the Expectation-Maximization algorithm. Before this work, there were two approaches to the problem of learning from corrupted data: Ambient Diffusion and SURE. Ambient Diffusion was designed for linear inverse problems and SURE for denoising. The proposed method provides a different, unified methodology for training diffusion models from corrupted data. The authors show experimentally that their method outperforms the Ambient Diffusion baseline in the same corruption setting.

**Strengths:**

1) The topic of learning diffusion models from corrupted data is very important and very interesting. The submission is timely and relevant as the interest in this research topic is growing.
2) The authors propose a fresh idea in the space of learning diffusion models from corrupted data. The proposed idea has several nice properties: i) it provides a unified treatment for different corruptions, ii) it is simple to implement, and iii) it has strong experimental performance.
3) The presentation of the paper is really good. The authors motivate the problem, present a new principled method to solve this, and show experimental results.
4) A byproduct of this work is a new method for solving inverse problems with diffusion models. While still approximate, this new method seems to be more principled compared to previous approaches.

Overall, this is a strong submission that offers new insights into the problem of learning from corrupted data.

**Weaknesses:**

The paper has also several weaknesses.

1) The proposed framework introduces significant computational and engineering overhead. Even for local convergence, EM typically requires many steps. In this setting, each step is a new training of a diffusion model.
2) The proposed algorithm is still approximate. The authors currently mention that their work is the first one that leads to "proper" denoisers. Yet, there is an approximation happening when the authors use diffusion models to perform posterior sampling. In fact, the authors had to develop a whole new method for solving inverse problems with diffusion models to get good results. The authors should emphasize this limitation more.
3) There is a very large field of prior works in solving inverse problems with diffusion models. This paper proposes a new method to achieve that goal. This method should be independently tested and benchmarked, decoupled by the context of learning with corrupted data. If the method is truly superior to prior works, this would be a very interesting byproduct of this paper and it's worth understanding this. If not, there is still value in the proposed method for the context of learning from corrupted data, but there is more to understand regarding why it works so well in this particular setting.
4) The authors currently provide comparisons with Ambient Diffusion only at the setting of $p=0.75$. It would be nice to see how their method scales for different corruption levels. It would also be helpful to provide comparisons in other datasets apart from CIFAR-10.
5) Since this method can be in principle used to train diffusion models from various inverse problems, it would be nice to see results for learning from other corruption types, e.g. blurry data.
6) Ambient Diffusion has been used to solve MRI inverse problems in the paper: "Ambient Diffusion Posterior Sampling: Solving Inverse Problems with Diffusion Models trained on Corrupted Data". It would be useful to provide comparisons with this work.
7) The authors do not provide comparisons for the additive Gaussian noise case. A natural baseline would be the SURE method. I would also like to bring the paper: "Consistent Diffusion Meets Tweedie" to the attention of the authors.
8) For some corruption types, it is impossible to ever reconstruct the true distribution from noisy observations. The authors do not discuss how the proposed algorithm would work in such cases.

**Questions:**

See weaknesses above. I would be happy to further increase my score if my concerns are properly addressed.

**Limitations:**

The authors have adequately addressed the limitations of their method.

---

> ### Author Rebuttal · Authors · 2024-08-05
>
> Thank you for your in-depth review and the legitimate concerns you have raised.
>
> * **W1** Indeed, this is one of the limitation of our method, which we mention in Section 3. However, we would like to mention that in our pipeline we start each training step with the previous parameters, which reduces the cost of each training step. Overall, the entire EM procedure for the corrupted CIFAR-10 experiment takes around 4 days for 32 EM iterations on 4 A100 GPUs (see Appendix C), which is similar to the training of AmbientDiffusion [77].
>
> * **W2** Our EM method is indeed sensitive to the quality of posterior samples, as mentioned in Sections 4.1, 4.2 and 7. We propose to add the following sentence at line 291.
>
>   > [...] sensitive to the quality of posterior samples. In fact, we find that previous posterior sampling methods [21, 22] lead to disappointing results, which motivates us to develop a better one.
>
> * **W3** We appreciate that you recognize MMPS as a valuable contribution in its own right. We follow your suggestion and benchmark MMPS against previous methods (DPS and $\Pi$GDM). We invite you to consult the global rebuttal regarding these additional experiments.
>
> * **W4** We follow your suggestion and repeat the corrupted CIFAR-10 experiment at more corruption levels (0.25, 0.50 and 0.75). We invite you to consult the global rebuttal regarding these additional experiments.
>
>   We note that the other experiments of AmbientDiffusion [77] also use a diagonal measurement $A$ (a mask), which is why we conduct the accelerted MRI experiment, where the measurement is a more challenging undersampled Fourier transform.
>
> * **W5** Indeed, a strengh of our method is that it can handle a wide variety of measurement types, or even learning from several sources with different measurement types. Our three experiments present different measurement types: a linear projection $A \in \mathbb{R}^{5 \times 2}$, a random binary masking, and an undersampled Fourier transform. We note that Gaussian blur, as you suggest, is equivalent to masking the high frequencies of an image, which is similar to the undersampled Fourier transform in the accelerated MRI experiment.
>
> * Concerning **W6** and **W7**, we thank you for bringing these concurrent works to our attention. We will discuss them in our related work section. Are we correct in understanding that in "Consistent Diffusion Meets Tweedie" the goal is to fine-tune a pre-trained diffusion model using data corrupted by isotropic Gaussian noise?
>
> * **W8** Indeed, we forgot to explain how our method would behave when $p(x)$ cannot be uniquely identified from the observations. We propose to replace lines 298-300 with the following paragraph
>
>   > Finally, as mentioned in Section 6, empirical Bayes is an ill-posed problem in that distinct prior distributions may result in the same distribution over observations. In other words, it is generally impossible to identify "the" ground-truth distribution $p(x)$ given an empirical distribution of observations $p(y)$. Instead, for a sufficiently expressive diffusion model, our EM method will eventually converge to a prior $q_\theta(x)$ that is consistent with $p(y)$, but generally different from $p(x)$. In future work, we would like to follow the maximum entropy principle, as advocated by Vetter et al. [36], so as not to reject any possible hypothesis.
>
>   We emphasize that this identifiability issue is a limitation of the problem itself, and not of our method.
>
> We believe that this rebuttal addresses most of your concerns and, therefore, kindly ask you to reconsider your score.

---

> > ### Comment · Reviewer_3yec · 2024-08-11
> > **Rebuttal acknowledgement**
> >
> > I would like to thank the authors for their efforts in the rebuttal. Indeed, most of my concerns are addressed. I highly encourage the authors to include this discussion and the additional experiments in the camera-ready version of their work.
> >
> > Regarding W4, please also include corruption levels above 0.75 in your camera-ready version and experiments on other datasets beyond CIFAR-10. It is crucial to have a holistic evaluation of the method in the same setting as prior work so that we can make progress and set up a nice benchmarking environment for future works. I believe this is very important for the field and I highly encourage the authors to include these additional experiments.
> >
> > Also regarding W4, for the MRI, I was not pointing to the Ambient Diffusion paper, but to the paper [Ambient Diffusion Posterior Sampling: Solving Inverse Problems with Diffusion Models Trained on Corrupted Data](https://arxiv.org/abs/2403.08728). This paper looks at exactly the same setting as yours and hence it should be cited and benchmarked.
> >
> > Regarding W6, Consistent Diffusion Meets Tweedie provides a general algorithm to train diffusion models from noisy data. In the context of the paper, it seems that this method is evaluated for fine-tuning a pre-trained model to a different dataset, but in principle, it should be doable to use it to train from scratch.
> >
> > I am increasing my score to 8. I hope that the authors will include the additional discussion and the experiments in their camera-ready and I am looking forward to reading this version.

---

> > > ### Author Response · Authors · 2024-08-13
> > >
> > > Thank you again for taking the time and effort to review our manuscript. Your constructive feedback and recognition of our work are deeply appreciated. We will make sure to include the additional discussion and experiments in the camera-ready version.

---

### Author Rebuttal · Authors · 2024-08-05

We would like to thank the reviewers for the quality and pertinence of their reviews. We are glad that all reviewers found the topic of our work interesting and timely.

Reviewers **3yec**, **XeV5**, **mKff** and **tnYw** found the method sound and well presented. Theirs concerns mainly regard the extent of the experiments. Notably, reviewers **3yec**, **mKff** and **tnYw** rightfully comment that the proposed Moment Matching Posterior Sampling (MMPS) method should be benchmarked independently from the context of learning from observations. We propose to address these concerns with additional experiments which we describe below.

Reviewer **s8XN** found the presentation of the method and its pipeline unclear, which prevented them from judging the contribution and results. We thank the reviewer for this opportunity to clarify our manuscript. We describe the relevant changes to the manuscript in reviewer **s8XN**'s rebuttal.

We would also like to emphasize that our work includes two contributions: a novel method to learn diffusion models from observations **and** an improved posterior sampling scheme. Many articles focusing on only one of these contributions have been published in major venues.

**Additional experiments**

* We **repeat the corrupted CIFAR-10 experiment at more corruption levels** (0.25, 0.50 and 0.75). We take the opportunity to refine the exponential moving average (EMA) decay rate of our training step to further improve the results of our method, which we present in Table 1 in the attached PDF. As expected, reducing the corruption level leads to even better final diffusion models. The EM algorithm also converges faster, which is expected as each observation $y$ conveys more information about its latent $x$. We propose to present and discuss these results in the main text.

* We **benchmark MMPS against SOTA posterior sampling methods**, namely DPS [21] and $\Pi$GDM [22], for several linear inverse problems on the FFHQ dataset. For a fair comparison, we adapt the official code published by Chung et al. [21] and use the provided pre-trained diffusion model as diffusion prior. We present premilinary results in the attached PDF. We consider 4 linear inverse problems:
    - Box inpainting with high noise ($\sigma_y = 1$)
    - Random inpainting with low noise ($\sigma_y = 10^{-2}$)
    - Motion deblur with moderate noise ($\sigma_y = 10^{-1}$)
    - Super resolution (4x) with moderate noise ($\sigma_y = 10^{-1}$)

    We find that MMPS requires very few sampling steps to generate qualitative samples and remains remarkably stable for challenging inverse problems (non-diagonal measurement and/or high noise). Conversely, DPS requires many steps to converge and $\Pi$GDM fails for moderate-to-high noise levels. We also find that increasing the number of CG iterations improves image quality/sharpness, but only marginally and with rapidly diminishing returns. Finally, our analysis (see Table 2) shows that an MMPS step is moderately slower (+16ms per CG iteration) than a DPS or $\Pi$GDM step, while only using 10% more memory. This hopefully addresses the concerns of high time and memory complexities raised by reviewers **tnYw** and **mKff**. We propose to present and discuss an extended version of these results in a new appendix section. We will also rewrite lines 291-297 in the discussion to better reflect these results.

---

### Decision · Program_Chairs · 2024-09-25

**Decision:**

Accept (poster)

**Comment:**

This paper considers a class of latent diffusion models where a latent variable $x$ is associated with every observation $y$, and is modeled as coming from a diffusion model $x \sim q_\theta(x)$ with parameters $\theta$. The paper proposes a Monte Carlo EM algorithm to learn $\theta$ wherein the E-step consists of sampling $x$ from the posterior $q_\theta(x \mid y)$ and the M-step consists of updating $\theta$ by denoising score matching. Due to the sensitivity of this procedure the on posterior samples in the E-step, the paper also proposes an improved method (denoted MMPS) for posterior sampling from diffusion models.

Most reviewers were excited by the paper, finding the methodological contributions timely, the results impressive, and writing to be of high quality.

There was initial concern among some reviewers about the computational  and memory overhead of the proposed procedure; however the authors responded convincingly to these concerns in their rebuttal. For example, reviewers were concerned that the EM algorithm essentially involved fitting a diffusion model from scratch every M-step. However, the authors clarified in the discussion period that the diffusion model is initialized at its last iterate, and further provided supplemental experimental evidence that indicated that the wall-clock time of the proposed method was on the same order as the baseline methods.

Reviewers also noted that the proposed posterior sampling method was interesting in its own right and deserved separate recognition and independent benchmarking. In response, the authors provided a suite of experimental results that compared MMPS to two previous methods, showing favorable results.

One reviewer also suggested additional comparisons to AmbientDiffusion with different hyperparameter values, which the authors supplied in the response, again showing favorable results.

There were a range of other questions from reviewers (e.g., regarding identifiability, relation to related work, and the impact of various approximations) to which the authors replied thoroughly and convincingly.